# Disruption of trait-environment relationships in African megafauna occurred in the middle Pleistocene

Daniel A. Lauer [1,2] ✉, A. Michelle Lawing[3], Rachel A. Short [4], Fredrick K. Manthi[5], Johannes Müller[6], Jason J. Head[7] & Jenny L. McGuire [1,2,8]

Mammalian megafauna have been critical to the functioning of Earth's biosphere for millions of years. However, since the Plio-Pleistocene, their biodiversity has declined concurrently with dramatic environmental change and hominin evolution. While these biodiversity declines are well-documented, their implications for the ecological function of megafaunal communities remain uncertain. Here, we adapt ecometric methods to evaluate whether the functional link between communities of herbivorous, eastern African megafauna and their environments (i.e., functional trait-environment relationships) was disrupted as biodiversity losses occurred over the past 7.4 Ma. Herbivore taxonomic and functional diversity began to decline during the Pliocene as open grassland habitats emerged, persisted, and expanded. In the mid-Pleistocene, grassland expansion intensified, and climates became more variable and arid. It was then that phylogenetic diversity declined, and the trait-environment relationships of herbivore communities shifted significantly. Our results divulge the varying implications of different losses in megafaunal biodiversity. Only the losses that occurred since the mid-Pleistocene were coincident with a disturbance to community ecological function. Prior diversity losses, conversely, occurred as the megafaunal species and trait pool narrowed towards those adapted to grassland environments.

Fewer than half of large-bodied mammalian genera that lived only 50,000 years ago are alive today[1]. Those that remain primarily live in Africa and Asia, but they are increasingly threatened with extinction as human impacts and climate change intensify[2–6]. For millions of years, these megafauna have been critical components of global ecosystems by structuring habitats[3,4,7,8], influencing fire regimes[3,9,10], cycling nutrients[4], serving as prey items[3,4], and carrying out many other essential functions[11,12]. To prevent the ecosystem disruptions that

would result from the continued loss of megafauna, we must understand how their ability to function in their natural environments has been threatened.

The relative dearth of megafauna on Earth today dates to their dramatic declines that began in the Pliocene and intensified towards the end of the Pleistocene. Globally, hundreds of the largest terrestrial species went extinct over the past 50,000 years[1,13,14]. In Africa, however, losses in megafaunal biodiversity are thought to have occurred earlier,

[1]Interdisciplinary Graduate Program in Quantitative Biosciences, Georgia Institute of Technology, Atlanta, GA 30332, USA. [2]School of Biological Sciences, Georgia Institute of Technology, Atlanta, GA 30332, USA. [3]Department of Ecology and Conservation Biology, Texas A&M University, College Station, TX 77843, USA. [4]Department of Natural Resource Management, South Dakota State University, Rapid City, SD 57703, USA. [5]Department of Earth Sciences, National Museums of Kenya, Nairobi, Kenya. [6]Leibniz Institute for Evolution and Biodiversity Science, Museum für Naturkunde Berlin, 10115 Berlin, Germany. [7]Department of Zoology and University Museum of Zoology, University of Cambridge, Cambridge CB2 3EJ, UK. [8]School of Earth and Atmospheric Sciences, Georgia Institute of Technology, Atlanta, GA 30332, USA. ✉e-mail: lauerd@gatech.edu

i.e., at multiple points since the early Pliocene[15–18]. Over this time period, megafauna experienced climatic changes[19,20], severe drought[14], and changes in vegetation across landscapes[10,13,15,16]. Simultaneously, hominins may have overhunted megafauna[8,21–31], spread disease to them[24], and/or encroached on their habitats and food sources[4,17,32], though these effects may have been mitigated in Africa (where hominins first evolved) by the long-term coevolution of megafauna with hominins[4,33,34]. The timing of megafaunal diversity losses, as well as their association with the emerging hominin clade and environmental change, has been heavily studied[1,8,13,14,16,24,31,35–40]. For example, Faith et al.[16] demonstrated that as grasslands expanded in eastern Africa, the richness of massive mammalian herbivores >1000 kg began a steady, long-term decline. While past diversity losses have been well-documented, their implications for the functioning of megafaunal communities have not been explored in depth.

Here, we investigate a unique facet of megafaunal biodiversity loss in Africa. We examine whether declines in taxonomic, phylogenetic, and functional diversity were coincident with disruptions in the functional relationships that megafauna have with their environments. To do so, we adapt an ecometrics approach. Ecometrics evaluates the relationships between functional traits of species making up communities and the environmental conditions of those communities[41–45]. Its theoretical basis is that certain suites of functional traits are better-suited to specific environments[41,46,47]. In mammalian herbivores, for example, a more durable (i.e., high-hypsodonty) tooth is well-suited to a more open environment comprising abrasive grit on grassy vegetation[41–43,48–52] and reduced woody cover[53]. When species possess traits well-suited to their environmental conditions, they can better leverage their traits to contribute to community function via survival, reproduction, and performance of their ecological role[48]. Communities composed of such well-suited species harbor strong trait-environment relationships[47,48].

While the goal of ecometrics is typically to reconstruct paleoclimate conditions[41–45,52], we adapt and develop it further by evaluating the consistency of trait-environment relationships over time and identifying if, and when, those relationships shifted. To do so, we establish these relationships using fossil and modern communities together, which provides us with a long-term baseline to determine when deviations in them occur. If ecometric relationships remain unaltered despite a loss in megafaunal diversity, then species are either tracking their preferred environments as they disperse, or they are undergoing trait changes with little evolutionary lag time as they adapt to new environmental conditions. But if a diversity loss is concurrent with a shift in ecometric relationships, then an interference may exist that alters the function of megafaunal communities with respect to their environments. In this way, ecometrics can distinguish between declines in megafaunal diversity that are results of habitat tracking and adaptation versus those that may disrupt community ecological function.

In this study, we evaluate if, and when, biodiversity losses were associated with disruptions to community function in large, eastern African mammalian herbivores (megafauna ≥44 kg), and we explore the roles that environmental change and hominin emergence may have played. We focus on the past 7.4 Ma, a period encompassing major environmental changes and the earliest occurrences of hominins[10,16]. Using a dataset of 203 fossil and 48 modern species of large herbivores, we first quantify temporal changes in herbivore taxonomic, phylogenetic, and functional diversity, each of which captures unique information[43,54,55]. We then use ecometric analyses (Supplementary Fig. 1) to evaluate communities of the large herbivores through time (Supplementary Fig. 2). For each of three functional traits (body mass, hypsodonty, and loph count), we build an ecometric model of trait-environment relationships[41–43,48,52] using data from the entire 7.4 Ma timespan to determine the most likely fraction of woody cover present at each community (Supplementary Figs. 1, 3). We then

calculate each community's ecometric anomaly, i.e., its measured woody cover fraction minus its most likely fraction[43,52,56] (Supplementary Fig. 1). Finally, we evaluate the consistency of trait-environment relationships through time by determining if, and when, community ecometric anomalies shifted. We find that such a shift occurred in the mid-Pleistocene, even though biodiversity began declining much earlier at ~5 Ma. This advances our understanding of past megafaunal diversity loss[1,14,16], as it suggests that the losses before versus after the mid-Pleistocene may have varied in their implications for the functioning of megafaunal communities.

## Results and discussion

### Megafaunal biodiversity decline and its environmental and paleoanthropological context

We observe that losses in the large herbivores' functional diversity commenced following the environmental changes of the late Miocene. Prior to ~8 Ma, woodlands and forests were prevalent across eastern Africa[57,58]. During the terminal Miocene before ~5 Ma, grasslands expanded[53,57,59–61] (Fig. 1, left-most blue region) and formed more open environments among the sites at which the large herbivores occurred (Supplementary Fig. 2). Habitats possessing ≤~40% woody cover (i.e., wooded grasslands[53]) persisted at these sites to 5 Ma (Fig. 1a). At that point, the trend in functional diversity (measured as the variation in species' body mass and dental trait values—see Methods; Supplementary Fig. 4) reached its breakpoint at 5 Ma (95% CI: 5.5–4.5) and began a sustained decline (Fig. 1d). The timing of this decline supports prior findings that the loss of taxonomic diversity in eastern African megaherbivores (>1000 kg), beginning 4.6 Ma, was associated with grassland expansion[10,15,16]. This decline also substantiates that since the late Miocene, environmental change induced shifts in functional diversity that engendered differences in the trait distributions of extinct versus modern herbivore communities[10]. Notably, the decline commenced at a time in which no major turning points in hominin evolution occurred[16,62].

Like the losses in functional diversity, subsequent losses in the large herbivores' taxonomic diversity also corresponded with environmental change. Leading up to the Mid-Pliocene Warm Period before 3.3 Ma[63] (Fig. 1, middle blue region), the trends in woody cover (Fig. 1a) and species richness (Fig. 1b) reached their maximum values. This was followed by respective breakpoints at 3.75 Ma (95% CI: 5.0–2.5) and 3.25 Ma (95% CI: 4.25–2.50) as both trends began a long-term decline. Both breakpoints preceded 3 Ma, when hominins developed basic Oldowan tools[64] (Fig. 1, middle orange bar). The decline in woody cover (Fig. 1a) was indicative of intensifying grassland expansion[10,16,53], a force that may have driven decreases in richness (Fig. 1b) by extirpating species less adapted to grass consumption[10,65]. Importantly, however, both trends did not decline synchronously, as richness increased steadily from 2.5 to shortly after 1 Ma (Fig. 1b). This suggests that other factors outside of grassland expansion may have driven the trend in richness as well.

Losses in phylogenetic diversity (measured as the degree of evolutionary divergence among species--see Methods) were the last to occur and were associated with a 200-Ky interval of importance to both environmental change and hominin evolution. The breakpoint in the phylogenetic diversity trend took place 2.25 Ma (95% CI: 2.75–2.00), but its first major decrease occurred during and following the interval of 1.9–1.7 Ma (circled points in Fig. 1c). ~1.9 Ma marked the emergence of the species *Homo erectus*[16,66]. *Homo erectus* represented a major turning point in hominin evolution[67], largely because of its development of relatively advanced Acheulean technology ~1.7 Ma[15,68] (Fig. 1, right-middle orange bars). Between 1.9 and 1.7 Ma, eastern African climates became more variable, and periods of aridity began to increase in frequency[67,69,70] (Fig. 1, right-middle blue region). Among the sites at which the large herbivores occurred (Supplementary Fig. 2), woody cover decreased to <35% (Fig. 1a). The increasing aridity

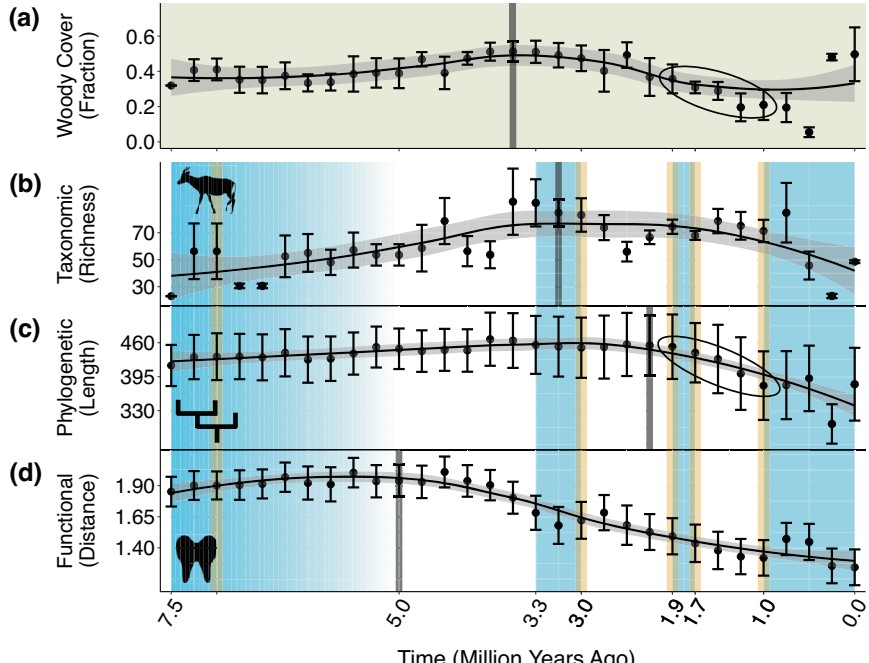

**Fig. 1 | LOESS regression trends in the environmental conditions and biodiversity of herbivorous megafauna in eastern Africa.** Trends show temporal changes in the fraction of woody cover among megafaunal communities (**a**), as well as in megafaunal taxonomic (**b**), phylogenetic (**c**), and functional diversity (**d**). Each data point represents a 250,000-year time bin. Among the species of megafauna occurring in each time bin, taxonomic diversity is measured as species richness; phylogenetic diversity as the sum of the branch lengths (in millions of years) of the phylogenetic tree connecting genera; and functional diversity as the mean Euclidean distance between species' trait values (body mass, hypsodonty, and loph count) and their centroid in three-dimensional space (see Supplementary Fig. 4). Each data point is presented as a mean value ± one standard error from $n = 1000$ independent samples of communities (**a**), species occurrences (**b**), genera (**c**), or species (**d**) (see Methods). LOESS regression curves use a smoothing parameter of 0.75. Faded gray bars denote breakpoints from breakpoint analysis. Blue-shaded areas refer to events related to environmental change and orange to events in hominin evolution. These events are encompassed in key time intervals, as follows: 7.5–5 Ma includes the onset of grassland expansion and the emergence of hominins (7 Ma); 3.3–3 Ma includes the mid-Pliocene Warm Period and the development of Oldowan hominin tools (3 Ma); 1.9–1.7 Ma includes the increase in climate variability and aridity, as well as the emergence of *Homo erectus* (1.9 Ma) and their development of Acheulean technology (1.7 Ma); and ≤1 Ma includes the intensification of periods of aridity, as well as rapid cranial growth in hominins (1 Ma). Circled points in **a** and **c** reference the most dramatic decline in phylogenetic diversity and the associated change in woody cover. Source data are provided as a Source Data file.

and openness of these environments could explain our observation that losses in phylogenetic diversity were particularly pronounced in non-ruminant species (Supplementary Fig. 5). Under arid conditions of limited vegetation, non-ruminants are at a disadvantage because they require more food to obtain the same amount of energy relative to ruminants[71,72].

## Biodiversity losses prior to mid-pleistocene associated with unaltered ecometric relationships

Our ecometric analyses reveal consistent trait-environment relationships throughout the Pliocene and early Pleistocene. Using a bootstrap resampling method, we determined when the ecometric anomalies of communities, with respect to their fractions of woody cover, differed significantly from zero through time. We analyzed anomalies within and between key time intervals that encompassed major events in both environmental change and hominin evolution: 7.4–5 Ma, 3.3–3 Ma, 1.9–1.7 Ma, and ≤1 Ma (Fig. 1; see Methods). Bootstrapped 95% confidence intervals contain zero from 7.4–5 Ma for body mass, as well as from 5 to 3.15 (midpoint of 3.3–3) Ma and 3.15–1.8 (midpoint of 1.9–1.7) Ma for all traits (Fig. 2, blue violin plots). This indicates that anomalies do not shift significantly across those periods. Consistent anomalies are also evident through the Pliocene and early Pleistocene when we consider the variation in woody cover within each community over time, when we use time bins that zoom in on 3.3–3 and 1.9–1.7 Ma, when we use time bins that are more uniform in size, and when we limit our analysis only to species ≥100 kg in mass (see Methods; Supplementary Figs. 6–10, blue violin plots; see also Supplementary Fig. 11).

Unaltered anomalies signify that while taxonomic and functional diversity were initially in decline (Fig. 1b, d), the trait-environment relationships of communities of the large herbivores did not change significantly. Species' abilities to leverage their body masses (Fig. 2a), hypsodonty (Fig. 2b), and loph counts (Fig. 2c) to contribute to community function remained undisturbed.

Therefore, the biodiversity changes that occurred before the mid-Pleistocene likely reflected that the large herbivores were tracking the emergence and persistence of open, grassland habitats[16,53,57] via movements across space or adaptations to new conditions. We find evidence of this starting before 5 Ma, when taxonomic (Fig. 1b), phylogenetic (Fig. 1c), and functional diversity (Fig. 1d) all initially increased. These increases were associated with the rise in large herbivorous grazers[50], which contributed new species, phylogenetic outgroups, and functional traits to the existing pool. Bovids ancestral to modern-day buffalo and antelopes, as well as members of novel genera like *Hippopotamus* were emerging. These taxa were well-adapted to the open grassland habitats that had become prevalent on the eastern African landscape[16,53,57,73,74], including within the sites at which they occurred (Fig. 1a; Supplementary Fig. 2).

## Ecometric relationships remained unaltered as species underwent trait changes

Specific functional trait changes in the large herbivores over time show how traits responded to the emergence and persistence of grasslands (Fig. 3; Supplementary Fig. 12). As functional diversity reached its peak 5 Ma (Fig. 1d), massive megaherbivores were common, contributing to

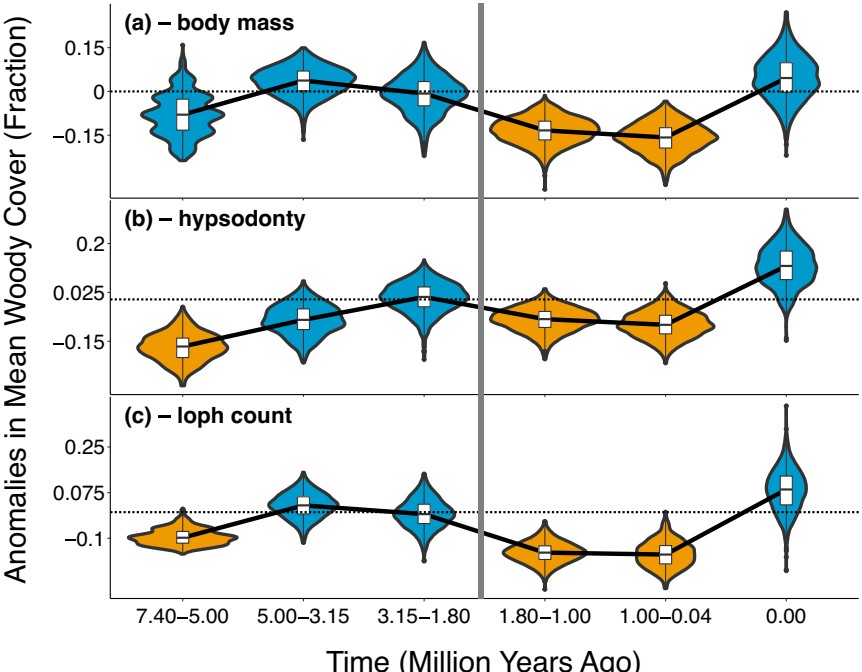

**Fig. 2 | Trends in the ecometric anomalies (see Supplementary Fig. 1d) of eastern African communities of large herbivores.** Anomalies are shown with respect to estimations of communities' mean woody cover values from herbivore body mass (**a**), hypsodonty (**b**), and longitudinal loph count (**c**). Maximum-likelihood estimations of woody cover were made for each community using an ecometric model of all communities together through time. Each violin plot depicts the distribution of $n = 1000$ independent samples, where each sample is the mean ecometric anomaly of a random subset of communities occurring within the plot's time bin (x axis). Each box plot depicts its distribution's median (center) and interquartile range (bounds of box), plus 1.5 * the interquartile range above and below the box (whiskers). Dotted lines indicate a mean anomaly of zero. Blue plots represent distributions whose 95% confidence intervals contain zero, while orange plots represent those whose confidence intervals do not. The vertical gray line represents the point at which ecometric anomalies shifted significantly after a long period of consistency. Time bins on the x axis are based on the cutoff points of the time intervals depicted in Fig. 1. 3.15 Ma is the midpoint of 3.3–3 Ma, 1.8 Ma is the midpoint of 1.9–1.7 Ma, and the final time bin represents the present. Source data are provided as a Source Data file.

an initially high mean body mass (Fig. 3a). Many of these mega-herbivores were browsers adapted to consuming softer, woody vegetation[10,15,16], possessing teeth with lower durability (low hypsodonty) and cutting ability (low loph count)[41–43,48,49] (Fig. 3c, e). Concurrently, large herbivorous grazers were emerging, producing initially high standard deviations in all traits (Fig. 3b, d, f) as the grazers added new traits to the existing pool. However, as wooded grasslands of ≤~40% woody cover[53] persisted among herbivore sites (Supplementary Fig. 2) past 5 Ma (Fig. 1a), and as grassland expansion intensified through the Plio-Pleistocene[16,73,74], a movement away from mega-herbivore browsers occurred[10,15,16]. Smaller browsers, which are less limited by forage availability[16], were at a competitive advantage as the supply of soft plant matter remained limited. This elicited a decline in the mean body mass of species (Fig. 3a). Grazers with durable and complex teeth were also at an advantage, causing mean hypsodonty (Fig. 3c) and loph count (Fig. 3e) to rise. For all traits, changes in mean were accompanied by decreases in standard deviation (Fig. 3b, d, f). Thus, functional diversity declined (Fig. 1d) as the existing trait pool narrowed in around traits suited for consuming relatively limited forage (lower body mass) or emerging grasses (greater hypsodonty and loph count).

The persistence of grasslands and the resulting functional trait changes of the large herbivores continued even through the mid-Pliocene Warm Period ~3.3–3 Ma[63] (Fig. 1, middle blue region). The early to mid-Pliocene was warmer and wetter than the more arid late Miocene, both globally[75] and across communities of eastern African herbivores[76]. While these wetter conditions increased woody cover[59] (specifically among the sites with herbivores (Supplementary Fig. 2)) to ~50% after 4 Ma (Fig. 1a), much of eastern Africa remained covered in grassland environments like tropical savannas[75]. Once woodlands were

sufficiently sparse by the end of the Miocene[53], wetter conditions favored increased grass cover[77] at the expense of more xeric ecosystems[75]. Further, the early Pliocene was characterized by the increased prevalence of fire[78], a force that promotes the emergence of grasses, but only if precipitation levels are sufficiently high[77]. Even during the wet mid-Pliocene, savannas were not overtaken by forests, as woody cover remained ≤~50% (Fig. 1a) and the traits of the large herbivores continued to track grassland environments (Figs. 2–3).

In two cases prior to the mid-Pleistocene, however, a lag between grassland expansion and herbivore trait change may have engendered an ecometric disruption. From 7.4–5 Ma, and unlike in the two periods that follow (5–3.15 and 3.15–1.8 Ma), the ecometric anomalies representing estimations of woody cover from hypsodonty and loph count differ significantly from zero (Fig. 2b, left-most orange violin plots). This reflects herbivore localities that comprised browsers adapted to the forested conditions of the earlier Miocene[58], despite exhibiting relatively low woody cover themselves (Fig. 1a). For instance, the community at Lower Nawata exhibited only 31.9% woody cover but included giraffid browsers like *Palaeotragus germaini*. The dental traits of species comprising such communities overpredict woody cover, resulting in ecometric anomalies that are significantly less than zero. These communities lagged in shifting their traits towards those better-suited to emerging wooded grasslands. Nonetheless, these lags occurred before the onset of biodiversity losses (Fig. 1). They do not detract from the broader pattern of consistent ecometric relationships as biodiversity declined prior to the mid-Pleistocene (Fig. 2; Supplementary Figs. 6–10).

Overall, we find that large herbivore diversity losses occurring before the mid-Pleistocene (Fig. 1b, d) were a result of species responses to environmental change, with no significant threat to

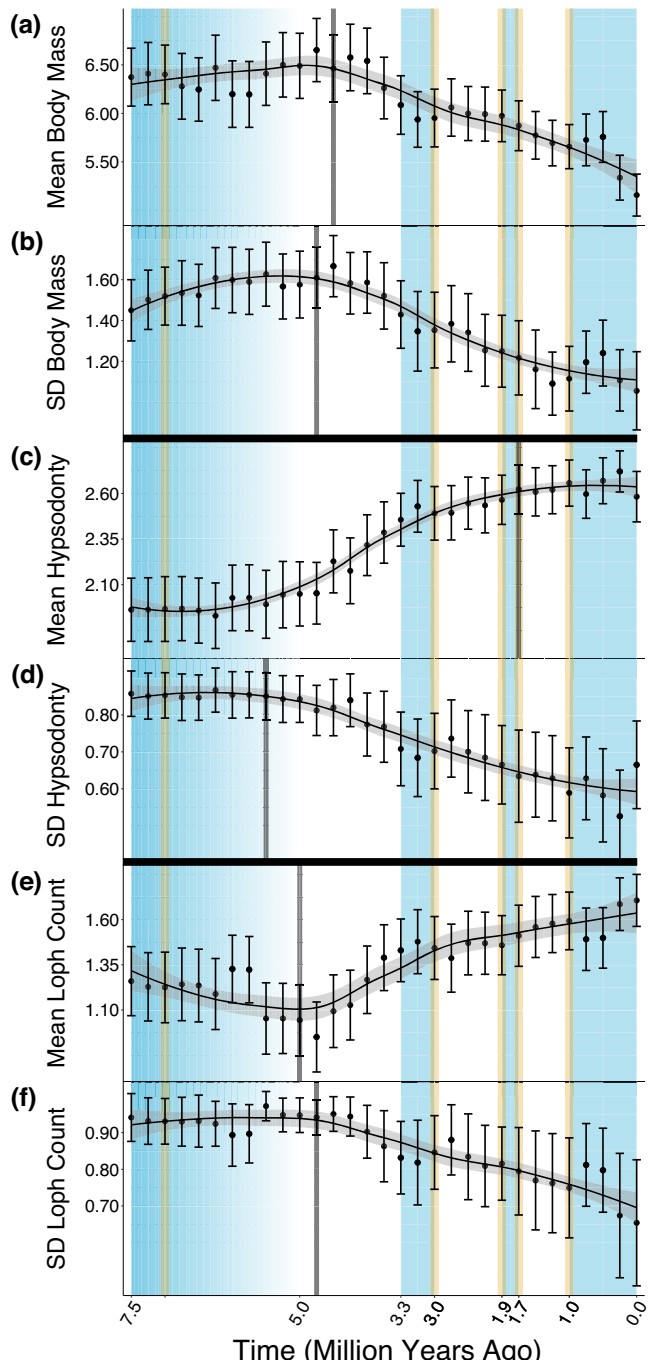

**Fig. 3 | LOESS regression trends in the mean and standard deviation of the trait values of eastern African large herbivores.** Trends show temporal changes in herbivore body mass (**a**, **b**), hypsodonty (**c**, **d**), and longitudinal loph count (**e**, **f**). Each data point represents a 250,000-year time bin. Body mass is measured in log-transformed kg, hypsodonty on a discrete scale of 1–3 (1 = brachydont, 2 = mesodont, 3 = hypsodont), and loph count as a count from 0 to 2. Each data point is presented as a mean value ± one standard error from n = 1000 independent samples of species (see Methods). LOESS regression curves use a smoothing parameter of 0.75. Faded gray bars denote breakpoints from breakpoint analysis. Blue-shaded areas refer to events related to environmental change and orange to events in hominin evolution. These events are encompassed in key time intervals, as follows: 7.5–5 Ma includes the onset of grassland expansion and the emergence of hominins (7 Ma); 3.3–3 Ma includes the mid-Pliocene Warm Period and the development of Oldowan hominin tools (3 Ma); 1.9–1.7 Ma includes the increase in climate variability and aridity, as well as the emergence of *Homo erectus* (1.9 Ma) and their development of Acheulean technology (1.7 Ma); and ≤1 Ma includes the intensification of periods of aridity, as well as rapid cranial growth in hominins (1 Ma). Source data are provided as a Source Data file.

direction, as ecometric anomalies fell significantly below zero. As in the case of hypsodonty and loph count from 7.4–5 Ma (Fig. 2b, left-most orange violin plots; see discussion above), these negative anomalies reflect communities whose component traits overpredict woody cover. More broadly, they signify that mid-Pleistocene biodiversity losses were distinct from those prior, in that they were associated with a disruption in the ecological function of those communities.

The mid-Pleistocene ecometric disruption, and the associated overpredictions of woody cover, could have resulted from changes in both biodiversity and local environmental conditions. Driving the decline in phylogenetic diversity (circled points in Fig. 1c) was the removal of genera whose species possessed traits adapted to grass consumption (e.g., high hypsodonty), like *Menelikia*. Their removal was at odds with the observed decrease in woody cover to <35% among herbivore communities (circled points in Fig. 1a) as grasslands expanded[16,73,74]. The loss of such genera, coupled with the decrease in woody cover, could have led to the assembly of communities with traits that were adapted to higher fractions of woody cover than were truly present. This would in turn explain our observation of an over-prediction of woody cover that characterizes the shift in trait-environment relationships (Fig. 2, orange violin plots). For the shift to have been avoided, the prevalence of traits adapted to grassland environments would have needed to increase sufficiently within herbivore communities to match the ongoing decrease in woody cover (Fig. 1a).

While our observation of the ecometric shift alone divulges the distinct implications of past biodiversity losses, it is noteworthy to explore what may have been the shift's underlying cause. Following 1.9 Ma, *Homo erectus* emerged on the eastern African landscape[16,66] and notable climate changes began occurring. Although it is hypothesized that *Homo erectus* impacted communities of large herbivores through hunting and carnivory[30,31,79], evidence of such behavior is not concrete[80] and hominins were still thought to live in relatively low densities[15]. Instead, the observed ecometric disruption may have been caused by the increase in the variability and aridity of eastern African climates. During and after 1.9–1.7 Ma, short periods of aridity were becoming more prevalent[67,69,70] (Fig. 1, right-middle blue region). They continued to occur until the present, and they grew in severity following 1 Ma[10] (Fig. 1, right-most blue region). These periods were decoupled from grassland expansion[16,81] and were particularly detrimental to grazers. Grazers often rely on drought-intolerant grass species for food[82], and such species would become scarce during dry phases. Grazers also cannot easily change their diets to adapt to the unique environmental conditions of drought periods[83]. Finally, because the water content of grasses is limited, grazers must depend

community ecological function. Our results support the hypothesis that declines in woody vegetation were associated with local mega-faunal diversity loss[10,13,15,16]. But while eastern African environments transformed[16,53,57,58,73,74], their communities of herbivorous megafauna maintained steady trait-environment relationships (Fig. 2, blue violin plots) as species continuously possessed functional traits well-suited to their environmental conditions.

### Biodiversity losses after mid-pleistocene coincident with ecometric disruption

Biodiversity losses that followed the interval of 1.9–1.7 Ma were associated with a disruption in trait-environment relationships. Across all three traits, herbivore communities experienced a significant shift in ecometric anomalies after that interval (Fig. 2; Supplementary Fig. 6–10, orange violin plots). This shift occurred in the negative

heavily on surface water for survival[84], but surface water retracts under arid conditions. Consequently, species adapted to grassland environments were at a disadvantage after 1.9–1.7 Ma, despite the continued expansion of grasslands at the sites in which they lived[10] (circled points in Fig. 1a). A long-term exclusion of such species, together with an increasing prevalence of grasses, would persistently engender herbivore communities whose traits suit environments that are woodier than were present. This provides a possible explanation for the observed long-term shift in trait-environment relationships, as communities would have been continually hindered from possessing traits adapted to open environments.

## Conclusions and future directions

While biodiversity losses in herbivorous eastern African megafauna have occurred for ~5 million years, the assembly and function of their communities were threatened only following those losses that occurred in the mid-Pleistocene. Prior losses resulted from large herbivores tracking the emergence, persistence, and expansion of grassland environments, with no threat to community ecological function. This is a key distinction that divulges the varying implications of different changes in biodiversity. These conclusions were only reachable once we combined complementary dimensions of diversity[85,86] with our adapted ecometric approach. Further, our conclusions add to those made by prior studies of Plio-Pleistocene megafaunal ecology. Prior research has suggested that since the mid-Pleistocene, environmental or anthropogenic forces may have driven changes in megafaunal biodiversity[10,17]. We take this a step further by demonstrating that environmental forces may have impacted megafaunal diversity in a way that was uniquely detrimental to their ecological function. Therefore, we urge future studies to incorporate ecometrics to better understand the consequences of changes in faunal biodiversity in the past, present, and future.

Moving forward, we must obtain a more complete picture of the ecological impacts of biodiversity losses stemming from anthropogenic activity and changing environments. Our work here focuses on eastern Africa, and only encompasses a subset of functional traits critical for functioning mammalian communities[87]. Future work should incorporate other traits and should include a wider diversity of taxa on a larger spatial scale. Traits like locomotion and taxa like carnivorans can provide critical insights into the complex interactions between megafaunal biodiversity, natural environments, and human impacts[47,88]. Only by understanding these interactions will we be able to ensure that megafauna can carry out their critical ecological functions in the years to come.

## Methods
### Data collection
**Occurrence and trait data.** Permissions or approvals were not required for this study. Our primary data source was a 2018 study documenting African megaherbivore extinctions over the past 7.4 Ma[16]. The data includes taxonomic classifications and occurrences of 349 mammalian herbivore species at 101 fossil sites, and 89 species at 203 modern sites. Species span the orders Artiodactyla, Perissodactyla, and Proboscidea. Their occurrences at fossil sites were compiled previously from literature and museum records, and at modern sites from records of national parks, game reserves, and protected areas[16,89,90]. The latter is more reliable than IUCN range maps, which may overestimate community composition[89,90]. Fossil sites span Ethiopia, Kenya, and Tanzania, ideal for our study because of eastern Africa's rich fossil record, relevance to hominin origins[16], and substantial environmental changes[58]. We only considered the 58 fossil sites that are associated with a known age range (i.e., the range of time in which a given site's component species occurred) and that have an available quantitative estimate of woody cover (see below for details). Modern sites span all of Africa to provide a robust baseline of current herbivore diversity[16] (Supplementary Fig. 2). As with the fossil sites, we

only considered the 128 modern sites that have available woody cover data.

We subset the data down to 203 fossil and 48 modern species. We removed fossil species with uncertain genus-level affiliations, e.g., a scientific name of *Gen sp* or a "?" preceding the genus name[91]. We also removed fossil species that occurred only at fossil sites without known age ranges or estimates of woody cover. We merged fossil species that do not refer to separate taxa, e.g., *Equus quagga* and *Equus* cf *quagga*[16]. However, we retained fossil species whose scientific names included "aff", e.g., *Eurygnathohippus hasumense* versus *Eurygnathohippus* aff *hasumense*, as "affinis" denotes a species that resembles, yet is certainly distinct from, another taxon[16,91]. To avoid double-counting fossil species' occurrences, we combined the occurrences of fossil species lacking a species-level identification with those of other species from the same genus[16]. After obtaining the body masses of the remaining species (see details below), we removed all those with a mass of <44 kg. Much of the literature addressing megafaunal extinctions recognizes megafauna as species weighing ≥44 kg[1,2,23,92]. Thus, the 44-kg threshold allows us to analyze megafaunal dynamics specifically and to be consistent with related literature. It is also a means by which to remove taphonomic biases, as small species are more difficult to discover in the fossil record[16].

To represent species' body masses, we collected the weight of each species in kilograms. We obtained body mass data primarily from Phylacine[93], PanTHERIA[94], and the NOW database[95]. However, if a species' weight was not recorded in one of the databases, we instead searched and acquired the missing data from the literature[16,96–124] (see Supplementary Data 1–2). We chose to analyze body mass because of its importance to functional diversity and ecometrics[50]. It is a trait that describes key properties of species: it determines the amount of heat that a species retains[125,126], the distance it travels, the strategy by which it defends itself[43], and the types of vegetation it can effectively consume[127].

We obtained data from the literature[40,44,50,122,128–136] to represent two more traits relevant to functional diversity and ecometrics: hypsodonty and longitudinal loph count. Together these traits illustrate a species' dentition and dietary adaptations. They influence which types of vegetation species are adapted to consuming, as different types of vegetation exhibit different levels of abrasiveness on teeth[49–51]. Hypsodonty is the ratio of a tooth's crown height to its width[49–51], discretized into values of 1 (brachydont), 2 (mesodont), or 3 (hypsodont)[95]. A lower value denotes that the entire tooth crown is exposed above the jawbone. Conversely, a higher value indicates that part of the crown remains in the jaw, emerging only as the exposed part of the crown is worn down by the processing of abrasive food[49,51]. Thus, a species with a higher hypsodonty value has a greater capacity for long-term tooth wear. Such a species has the durability to chew through the rough grit and dust that accumulates on grassy vegetation in open environments[49–51,137,138]. Loph count denotes the mean number of ridged structures on a tooth, discretized into values of 0, 1, or 2[50]. A species with a higher loph count value has more ridged surface area on its teeth, giving it the cutting ability to consume gritty grasses[136]. Although hypsodonty and loph count are globally correlated, they are each still distinctly important. Hypsodonty has a more varied global distribution and emphasizes tooth durability[139], while loph count emphasizes tooth-cutting ability and complexity[50]. Further, both are important to consider in addition to body mass. While a species' dentition and size are often related[140], these dental traits scale isometrically with body mass[49,51,141]. Thus, body mass is not necessarily a fully reliable proxy for hypsodonty and loph count in every context.

For the 45 taxa lacking a species-level identification (e.g., *Aepyceros* sp), we estimated their body mass, hypsodonty, and loph count values from those of closely related species. Confident determinations of such species' hypsodonty and loph count values were made directly from the literature, based on the values associated with other closely

related members of the same genus. In the 43 cases where confident literature determinations could not be made for a species' body mass, its mass was instead calculated as the mean mass of other members of its same genus (see Supplementary Data 1 for details). This allowed us to perform all trait-based analyses at the species level.

**Woody cover data for ecometrics.** For the ecometrics component of our study, we collected the woody cover data necessary to assess species' trait-environment relationships over the past 7.4 Ma. Woody cover refers to the fraction of a site's area that is covered by tree canopy. For 58 of the fossil sites, data were available to estimate woody cover from soil carbon isotope measurements (primarily from Levin[142], but see Supplementary Data 3[53,142–148]). Cerling et al.[53]. developed a procedure and equation to convert such measurements to values of woody cover. Thus, for each of these 58 sites, we performed the following: (1) we obtained all isotope measurements that were taken at the site and that were dated to an age that fell within the site's age range (see Supplementary Data 3 for age ranges from Faith et al.[16]); and (2) we converted each of these measurements to a woody cover value. Many sites were associated with more than one measurement, often representing different values of woody cover for different points in time throughout a site's age range. In those cases, we calculated each site's mean and standard deviation of woody cover across all its measurements. For 128 of the modern sites, woody cover data were directly available and obtained from Barr & Biernat[149] (see Supplementary Data 4).

## Data analysis
**Dimensions of biodiversity analysis.** We analyzed how the biodiversity of the herbivorous megafauna in our dataset changed over the past 7.4 Ma and compared those changes to the timing of past events related to environmental change and hominin evolution. While our time series analysis was modeled after Faith et al.[16], we adopted a different methodological approach based on differences in our study focus. Faith et al.[16] used a residual-based method to estimate the richness of megaherbivores (herbivores weighing >1000 kg) over time as follows: (1) linearly modeling megaherbivore richness as a function of overall herbivore community richness across all modern sites; and (2) plotting the residuals of all fossil sites in that model as a function of the age at which those sites occurred. This method is not appropriate in our case, as we analyze not just megaherbivores, but all herbivorous megafauna (herbivores weighing ≥44 kg). This allows us to investigate a wider array of taxa, taxa that are a major focus in the literature addressing Plio-Pleistocene megafaunal biodiversity loss[1,2,23,92]. Unlike megaherbivores, herbivorous megafaunal species make up the vast majority (>80%) of all species in our processed dataset. Thus, employing a model like in step 1 above would be uninformative for our analyses.

In examining megafaunal biodiversity, we considered three of its dimensions, namely taxonomic, phylogenetic, and functional diversity. Each captures unique information about how biodiversity may have responded to past events. Changes in taxonomic diversity signals if an event was followed by an increase or decrease in the number of species extant. Changes in phylogenetic diversity indicates if an event possibly led to the emergence of novel evolutionary lineages, as well as the continuation or disappearance of established ones[54,55]. And changes in functional diversity specifies how an event may have altered the prevalence of species' functional traits, traits that dictate their interactions with their surroundings[43]. Thus, taxonomic diversity represents how many taxa, phylogenetic diversity which taxa, and functional diversity which types of taxa, forming a spectrum of broader to more nuanced depictions of diversity.

To analyze these three dimensions of diversity, we first considered that each of the fossil sites represents a range of time. This allowed us to determine the minimum and maximum age through

which each fossil species existed in our study area: for each fossil species, we noted the temporal ranges of each of the sites at which it occurred, and then recorded the earliest and latest points in time represented across those ranges. We then used the knowledge of when each species did and did not occur to examine temporal patterns in each dimension of diversity. Once these patterns were identified, we overlaid them on a series of time intervals that each encompassed pivotal events related to both environmental change and hominin evolution. The intervals, and their component events, were as follows:

1. 7.4–5 Ma: this interval encompassed the onset of grassland expansion[53,57,59–61] and the evolution of the earliest known hominins[150].
2. 3.3–3 Ma: this interval encompassed the mid-Pliocene Warm Period[63] and the development of basic hominin Oldowan tools[64].
3. 1.9–1.7 Ma: this interval encompassed an increase in the variability and aridity of African climates[67,69,70], as well as the emergence of *Homo erectus* and their development of relatively advanced Acheulean tools[15,66,68].
4. ≤1 Ma: this interval encompassed the intensified frequency and severity of periods of aridity[10], as well as the rapid acceleration of cranial growth in hominins[62].

To detect temporal patterns in diversity, we identified the species occurring within each of a series of time bins and measured the diversity of the species in each bin. We chose to employ sequential 250,000-year time bins (7.5–7.25, 7.25–7, ..., 0.25–0 Ma). 250,000 years provides a proper balance between temporal resolutions that are too coarse or fine. It is coarse enough to reduce Signor-Lipps effects (i.e., when a species' true age of origination is offset from its oldest recorded age in our dataset[151]), because a coarser bin increases the chances that species' true origination dates and oldest recorded ages occur in the same bin. Simultaneously, it is fine-scale enough to capture changes in diversity across million-year ranges of time while preventing any single time bin from having too much of an effect on observed trends. Nonetheless, we ran a sensitivity analysis to determine if changing the time bin size and using a moving window (7.5–7, 7.25–6.75, ...) impacted our results. We found that in every case, the overall shapes of the trends observed in each of taxonomic, phylogenetic, and functional diversity, as well as in woody cover (see below for details), were highly similar (Supplementary Fig. 13). Per time bin, we measured taxonomic diversity as species richness. We measured phylogenetic diversity using methods described in detail in the *Phylogenetic Diversity* section below. Finally, we measured functional diversity in two ways: 1) by plotting the body mass, hypsodonty, and loph count values of all species in the time bin in three-dimensional space, and then determining the mean distance between each plotted species and the centroid of all plotted species[152,153]; and 2) by computing the mean and standard deviation of each trait across the species in the time bin. We repeated the latter across all species prior to applying the 44-kg threshold, as dental trait trends may be artificially affected by our removal of species <44 kg in mass. We found that trends that include versus exclude species weighing <44 kg were highly similar (Fig. 3; Supplementary Fig. 12).

In measuring each dimension of diversity over time, we accounted for the temporal sampling bias associated with the lower levels of sampling of fossils from older sites. For taxonomic diversity, we did so using coverage-based rarefaction[154], a more robust estimator of species richness in the fossil record than other popular methods[155]. Broadly, coverage-based rarefaction performs the following for each of our 250,000-year time bins: (1) subsamples a group of occurrences of species from across the fossil sites in the time bin; (2) identifies the species represented in that subsample and counts how many there are; and (3) determines what percent of all occurrences in the time bin's fossil sites are members of the species identified in step 2 (i.e., the coverage of the occurrence subsample). Then, coverage-based

rarefaction can ensure that calculations of species richness within each time bin use an equal percentage of coverage for all bins[156,157]. This method produces a standard error for the estimate of richness in each time bin across 1000 occurrence subsamples. As such, we plotted each estimate with error bars representing one standard error above and below it. For phylogenetic and functional diversity, we accounted for temporal sampling bias via bootstrap resampling. Within each time bin, we randomly sampled a subset of taxa with replacement 1000 times, the subset being equal in size to the absolute number of taxa occurring in the oldest time bin. We then performed the diversity calculation for each sample and recorded the mean calculation across all 1000 samples[158]. We also recorded the standard deviation of the 1000 calculations (i.e., the standard error), and we plotted each mean value with error bars representing one standard error above and below it.

We performed the same bootstrap resampling method as in our analysis of phylogenetic and functional diversity (see paragraph above) to produce a parallel trend in woody cover. The main difference here was that instead of randomly sampling subsets of taxa, we instead sampled subsets of sites whose age ranges overlapped each time bin. Further, we set the size of each subset to be equal to the absolute number of sites occurring in the oldest time bin that contained more than one site, as the oldest time bin (7.5–7.25 Ma) only contained a single site.

Once the dimensions of diversity and woody cover values were calculated for each time bin, we analyzed their resulting trends in two ways. For each type of calculation performed, we first plotted the calculated values as a function of time and fit a LOESS regression curve to the plot with a smoothing parameter of 0.75. LOESS regression curves are valuable for providing a visual of the temporal patterns of the values, including how patterns may change after past events. Then, using the 'segmented' package in R[159], we performed breakpoint analysis on the plotted values. Breakpoint analysis finds the point along the time axis that minimizes the error of piecewise linear regression lines fit to the calculated values before and after it. Thus, it signals the point in time at which a trend most dramatically changes direction, a point that can be equated to the timing of past events. We performed a Davies test[160] on each of our trends to determine if such a breakpoint is indeed appropriate. The Davies test can determine whether a trend is better fit using a single regression line, or alternatively using two regression lines that are separated by a breakpoint. For all trends, we determined that the latter option led to a better fit ($p < 0.05$), and therefore that a breakpoint is appropriate. In using the *segmented* function for breakpoint analysis, we increased the number of its default bootstrap resampling iterations up to 1000. This is in accordance with prior analyses of long-term patterns in species dynamics[161].

**Dimensions of biodiversity analysis—phylogenetic diversity.** To measure phylogenetic diversity, we obtained a set of phylogenetic trees from the VertLife Project (http://vertlife.org/phylosubsets/)[162]. VertLife maintains sets of trees that encompass 5911 extant and recently extinct mammalian species. Together, these trees capture uncertainty in the topography and temporal placement of the root, nodes, and tips describing the evolutionary relationships between mammals through time. The trees were constructed and dated by Upham et al.[162]. primarily by applying Bayesian inference methods to known DNA sequences of species. Species without DNA data were imputed into the trees based on their taxonomic classifications (see details below). Each branch in all trees was assigned a length, in millions of years, representing the amount of time that has passed between the nodes or node-tip combination making up the ends of the branch[162]. Thus, a group of taxa that are more closely related would be connected to each other by fewer and/or shorter branches, and vice versa.

Using the 'phytools' and 'ape' packages in R[163,164], we processed 100 full, randomly selected phylogenetic trees downloaded from the VertLife Project website. We pruned each tree down to only the tips of the species within the genera making up the fossil and modern large herbivores in our dataset. Because of the lack of species-level identifications for a subset of the fossil herbivores, the timing of the evolutionary trajectories of those herbivores past the genus level is unknown. We, therefore, conducted our analysis of phylogenetic diversity at the genus level. We note, however, that we could nonetheless perform our other analyses at the species level. For taxonomic diversity, we only required knowledge that species in an assemblage were distinct, which could be accomplished without knowledge of each's species-level identification. And for functional diversity and ecometrics (see below), we estimated all species' trait values (see *Occurrence and Trait Data* above). To convert our phylogenetic analyses to the genus level, we retained, for each of the 100 trees, the tip of only one random species from each genus. We note that the choice of which random species was retained per genus did not affect our results, because we measured phylogenetic diversity each time for a series of genera that were all concurrent (see details below). Thus, when each tree was pruned to a group of concurrent genera, its tips, each representing a genus, were aligned (i.e., each tree was ultrametric). The lengths of the tips were then set, regardless of the initial species chosen to represent each genus.

The trees from the VertLife Project and other data sources[165] only encompass around half of the 73 genera in our dataset, necessitating an imputation to include missing genera. Per the guidance of Upham et al.[162], we performed the imputation on each of the 100 trees as follows:

1. For each missing genus, we obtained its taxonomic classification.
2. Among the non-missing genera already in the tree, we determined which of them share the same Tribe, or otherwise Subfamily, or otherwise Family, or otherwise Order as the missing genus. Thus, we searched for taxonomic matches first among the narrowest levels of taxonomic classification, progressing to coarser levels only if necessary.
3. We located the most recent common ancestor node of the genera, or alternatively the tip of the singular genus, sharing a match with the missing genus.
4. We added the missing genus to the tree as a branch emerging out of the common ancestor node, or as a branch sharing a node with the singular genus. In this process, the length of the new branch was scaled such that its tip would align with the other tips in the tree, as the tips in the original tree obtained from VertLife were already aligned. The alignment of branches does not affect our results, because we measured phylogenetic diversity within each time bin for a series of simultaneously extant genera. Consequently, a tree that is pruned to such concurrent genera would be ultrametric, i.e., composed of tips that align with each other, regardless of the initial tree's topography.
5. Step 4 introduces polytomies to the tree, as a given node may now have more than two downstream branches. The polytomies represent the uncertainty surrounding the order in which the missing genus diverged relative to the non-missing genera. We resolved them by converting each polytomous node into a series of dichotomous nodes, where the order of divergence among the genera in the polytomous node is randomized. By repeating this multi-step process across the 100 trees, we account for a variety of possible evolutionary trajectories among the 73 large herbivorous genera and thus boost the robustness of our results. In fact, a sensitivity analysis revealed that when we performed multiple random iterations of this five-step imputation process, followed each time by all subsequent analyses, the trend observed in phylogenetic diversity remained the same each time.

Having generated 100 trees that now each contain all 73 genera in our dataset, we were able to measure phylogenetic diversity for each 250,000-year time bin employed in our analysis of large herbivore biodiversity. Per time bin, we pruned each of the 100 trees such that they only encompassed the genera occurring in the bin. Then, per pruned tree, we calculated the sum of the lengths of all of its branches, i.e., all branches extending from the root to the tips[54]. Finally, we computed the mean of those summations across all 100 trees. In this way, we computed a single mean phylogenetic diversity value for all time bins. We repeated this measurement of phylogenetic diversity over time separately for ruminant (i.e., bovids) and non-ruminant genera, to account for their differential diversity patterns over the period of time in which phylogenetic diversity was in decline[10] (Fig. 1c).

**Ecometrics analysis.** Our ecometrics analysis examines how the trait-environment relationships of the large herbivores in our dataset changed over the past 7.4 Ma. An ecometrics analysis investigates how the functional traits of species in communities relate to communities' local environmental conditions[41–45]. It is performed at the community-level, which allows for the discovery of patterns that would be subject to noise in populations or individuals[42,139]. By performing such an analysis across all fossil and modern communities of large herbivores, we were able to determine if and how the herbivores' trait-environment relationships were altered through time.

In our ecometric approach, we evaluated the relationship of community-level body mass, hypsodonty, and loph count with the fractions of woody cover that occur within communities. We considered woody cover as our environmental variable for several reasons: (1) it is a strong proxy of the composition of vegetation across eastern African sites, as it signifies the degree to which sites are composed of trees or grasses;[53,166] (2) it is a more reliable representative of late-Cenozoic grassland expansion in eastern Africa than other environmental variables, like temperature and precipitation;[81,167] and (3) the woody cover data at the fossil and modern sites in our dataset are of high quality[142,149]. Body mass is fundamentally related to woody cover because of its role in optimizing nutrient acquisition from vegetation types of different nutritive quality[127]. In Plio-Pleistocene eastern African environments, greater woody cover occurred when the quality of that woody vegetation was lower[16]. A greater body mass was advantageous under such conditions, as larger-bodied species can digest food for longer periods to extract nutrients from within low-quality forage, and they require fewer nutrients per unit of body weight[127]. Conversely, lower woody cover occurred when vegetation was higher in quality. A lower body mass was then more beneficial, as smaller species can persist from such vegetation in smaller absolute amounts than their larger counterparts[16]. Hypsodonty and loph count are fundamentally linked to woody cover as well. Regions of limited woody cover are typically composed of open habitats with abundant grassy vegetation[53]. Because grasses generally occur in open, exposed areas, they tend to accumulate soil and grit. Thus, grassland environments are more fitting for grazers with high-hypsodonty and high-loph-count teeth that can durably wear down and cut through rough, gritty material[41–43,48–51,138]. Conversely, regions of greater woody cover are more abundant in soft plant matter that is suitable for browsers with less durable dentition.

To begin our ecometric analysis, we assessed the traits of the large herbivore species occurring at each fossil and modern site. Per site, we calculated the mean and standard deviation of the body mass, hypsodonty, and loph count values of its component species. We only considered sites that consist of at least three species. A threshold of three species has been used in prior ecometric research[56]. It excludes sites that contain too few species for calculating meaningful mean and standard deviation measurements, while otherwise maximizing the number of sites included in our analysis. Importantly, a recent study cautions that such measurements may be biased by sampling effort[44]. In accordance with the study's guidelines, we tested for bias by

modeling the relationship between each mean or standard deviation measurement and sites' taxonomic richness values across all sites. We found that taxonomic richness explains minimal percentages of the variance in each measurement (mean body mass: 15.6%, mean hypsodonty: <0.1%, mean loph count: <0.1%, standard deviation body mass: <0.1%, standard deviation hypsodonty: <0.1%, standard deviation loph count: <0.1%). Thus, we may confidently assume that these measurements are not substantially influenced by sampling bias[44]. In addition, we also find that these measurements are not substantially biased by time-averaging. A weak correlation exists between taxonomic richness and the magnitudes of age ranges associated with fossil sites ($r = -0.01$). This means that sites with constrained age ranges do not have biased trait distributions relative to sites with broader age ranges, supporting the use of these sites in community-level analyses[10,16].

We built the ecometric models that follow using a series of repetitions that accounts for time-averaging in measures of woody cover at the 58 fossil sites. Specifically, we accounted for the fact that across the age range of each fossil site, woody cover likely varied over time. We did so by repeating all ecometric analyses using not only the mean woody cover values of the fossil sites, but also using the values representing one standard deviation above and, separately, below their means. Notably, these standard deviations were weakly related to, and thus were not biased by, the number of carbon isotope measurements taken at each site ($r = 0.18$).

We built an ecometric model separately for each trait. We did so first by organizing all sites into a two-dimensional grid of trait bins based on their component species' traits. Each site was binned based on its mean ($x$ axis) and standard deviation ($y$ axis) trait values for the trait in question (see Supplementary Fig. 1b). To determine the number of trait bins employed per axis, we used the Scott method, which determines the optimal number of bins for a distribution of values based on those values' standard error[168]. The Scott method ensures that each trait bin has the greatest sample size of component sites, boosting the statistical power of our analysis.

For each trait, and for each trait bin containing more than one site, we calculated the maximum-likelihood fraction of woody cover of the sites in the bin (Supplementary Fig. 3). We did so by producing a Gaussian probability density function (PDF) from the woody cover values of the trait bin's sites. The peak of the function is then the maximum-likelihood value for all sites in the bin (see Supplementary Fig. 1c). This maximum-likelihood approach has been demonstrated to be the most effective strategy for estimating environmental conditions from species' traits[52]. We performed it per trait bin as follows, to account for the temporal sampling bias associated with the greater number of sites occurring at more recent time points:

1. We calculated the mean of the temporal range of each fossil site within the trait bin, or otherwise designated each modern site as modern.
2. We grouped each fossil site by whether the mean of its temporal range falls into 7.4–5, 5–3.15 (midpoint of 3.3–3), 3.15–1.8 (midpoint of 1.9–1.7), 1.8–1, or 1–0.035 (youngest fossil site age) Ma. We chose these time bins based on the cutoff points of the key time intervals listed in the *Dimensions of Biodiversity Analysis* section above (see Fig. 1).
3. We assigned a sampling probability to each fossil and modern site in the trait bin based on the time bin in which it falls, ensuring that each time bin had an equal total sampling probability. For example, consider a trait bin that contains one site whose temporal range mean falls between 7.4–5 Ma, two sites 5–3.15 Ma, one site 1.8–1 Ma, and four modern sites. That trait bin's site-associated sampling probabilities would be, in order, 0.25, 0.125, 0.125, 0.25, 0.0625, 0.0625, 0.0625, and 0.0625. This would ensure that each time bin, plus the modern time bin, would each encompass a site sampling probability of 0.25 and could each be sampled from with equal frequency.

4.  We bootstrap resampled a subset of random sites from the trait bin with replacement, weighting the resampling by the probabilities determined in step 3.
5.  We created a Gaussian PDF from those random sites' woody cover values and found the peak of the PDF. That peak is the maximum-likelihood value.
6.  We repeated steps 4 and 5 1000 times, and we calculated the mean of the 1000 PDF peak values. That mean is the final maximum-likelihood value for the trait bin.

To assess the degree to which these maximum-likelihood estimations effectively capture trait-environment relationships, and to perform further analyses, we used ecometric anomalies. For each ecometric model built, we refer to each fossil and modern site's ecometric anomaly as its measured minus maximum-likelihood-estimated woody cover value (see Supplementary Fig. 1d). Thus, each site was associated with such an anomaly per trait. For each ecometric model (one for each trait), we plotted its distribution of ecometric anomalies across all sites and assessed the degree to which those anomalies surround zero (see Supplementary Fig. 11). Distributions of anomalies that more tightly surround zero (as opposed to being uniform in shape) denote maximum-likelihood estimations that reliably capture ecometric relationships[52].

We also used ecometric anomalies to evaluate the consistency of trait-environment relationships across past events. We aimed to determine the degree to which anomalies, and thus trait-environment relationships, changed over time. This could be done by comparing the ecometric anomalies of the sites that comprise successive time bins. As in step 2 above, these time bins were structured based on the intervals introduced in the *Dimensions of Biodiversity Analysis* section above, and included the present day (7.4–5, 5–3.15, 3.15–1.8, 1.8–1, 1–0.035, 0 Ma). For each time bin, we performed the following steps 1000 times to conduct our analysis while accounting for temporal sampling bias:

1.  We bootstrap resampled a random subset of sites whose age range falls into the time bin (for the time bins that are not 0 Ma) or which are modern (for the 0-Ma time bin). We set the size of the subset to be equal to the number of sites falling into the 7.4–5 Ma range, which contains the least number of sites relative to the other time bins.
2.  We calculated the mean of the ecometric anomalies of that random subset of sites.

Thus, per time bin, we produced a distribution of mean ecometric anomalies. If, for a series of successive time bins, the 95% confidence intervals of their distributions contain zero, then the ecometric anomalies, and therefore trait-environment relationships, across those time bins has remained consistent. The opposite is true if a time bin's confidence interval does not contain zero, as that would signify a shift in ecometric relationships.

In addition to repeating our ecometric analysis using the mean and standard deviation woody cover values of the 58 fossil sites (see above), we performed three other repetitions of the analysis. First, we repeated our approach using time bins that focus in on the intervals 3.3–3 and 1.9–1.7 Ma (7.4–5, 5–3.3, 3.3–3, 3–1.9, 1.9–1.7, 1.7–1, 1–0.035, 0 Ma). Second, we repeated it using time bins that are independent of past events and more uniform in size. Using time bins that are identical in size, while also employing a meaningful number of time bins, was a challenge considering the highly limited sample of sites that occur in earlier times (for example, only 8 sites occur across the first 3 Ma of our time series). If we used time bins that are too small, particularly for earlier times, those bins would encompass too few sites to produce meaningful distributions of ecometric anomalies. Thus, we used two 2-Ma time bins, followed by two slightly smaller 1.7-Ma bins (7.4–5.4, 5.4–3.4, 3.4–1.7, 1.7–0.035, 0 Ma). Finally, we repeated our analysis

when only considering species that are ≥100 kg in mass. 100 kg is another threshold that has been used to identify herbivorous mammalian megafauna[2,169]. By considering it, we were able to show that our results are not specific to only our chosen 44-kg threshold.

We wish to emphasize two components of our ecometric analyses, which can serve as a framework for future evaluations of ecometric relationships. First, we chose to focus not on the ecometric anomalies of sites at face value, but instead on how they changed over time. We found this to be a meaningful way to analyze temporal patterns in trait-environment relationships. Second, we chose to build and analyze ecometric models of all fossil and modern sites from all time bins together. If we instead performed our analysis on the sites in each time bin separately, we would be unable to effectively examine how ecometric anomalies changed across sites over time, since the ecometric relationship for each time bin would be separate and not comparable. Further, in building our ecometric models with all sites at once, we chose not to split sites into training and testing subgroups. This maximizes the sample size of sites used to determine maximum-likelihood climates. Prior research has shown that employing a train/test split leads to minimal differences in the ecometric results observed[56]. We performed all analyses for this study in RStudio version 3.6.1[170].

### Reporting summary

Further information on research design is available in the Nature Portfolio Reporting Summary linked to this article.

## Data availability

The data generated in this study are provided in the Supplementary Information and Source Data file. The Supplementary Information includes Supplementary Data 1–4, which are available as Microsoft Excel files. The legends for these datasets, which describe the datasets in detail, can be found in the Description of Additional Supplementary Files file. Source data are provided with this paper.

## Code availability

The R code written to conduct all analyses, and the files on which that code depends, are available on GitHub at https://github.com/lauerd/MegafaunaEcometrics (https://doi.org/10.5281/zenodo.8019076)[171].

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

## Acknowledgements

We thank J.T. Faith for helpful input on the manuscript. We also thank N. Upham for direction on how to obtain and process the phylogenetic trees analyzed in our study, and S. Holland and J. Alroy for consultation regarding species richness estimation. Conservation Paleobiology in Africa (CPiA) and Integrative Climate Change Biology (iCCB), which was funded by the International Union of Biological Sciences (IUBS), provided the framework to develop the initial research idea and to bring the co-authors into collaboration. The Spatial Ecology and Paleontology Lab (SEPL) at the Georgia Institute of Technology provided the intellectual environment necessary to complete the present research. This work was completed as part of a collaborative initiative from NSFDEB-NERC, with funding from NSF # 2124836 to A.M.L., F.K.M., and J.M., NSF # 2124770 to J.L.M., and NERC NE/W007576/1 to J.J.H. R.A.S. was supported by the NSF Postdoctoral Research Fellowships in Biology Program under Grant # DBI 2010680 and the USDA NIFA Hatch project SD00H787-23 (7004129 and 7004187). J.L.M was also funded through NSF-CAREER, NSF #1945013.

## Author contributions

D.A.L. carried out all data analyses and wrote the manuscript. A.M.L., F.K.M., J.M., J.J.H., and J.L.M. devised the initial research idea, and J.L.M., A.M.L., J.J.H., R.A.S., and J.M. provided continuous guidance and editorial support. R.A.S. compiled and contributed some data.

## Competing interests

The authors declare no competing interests.
