## [Peer Review File · Nature Communications]

Disruption of trait-environment relationships in African megafauna occurred in the middle PleistoceneReviewers' Comments:

Reviewer #1:

Remarks to the Author:

This paper addresses biodiversity loss of eastern Africa megafauna through the Plio-Pleistocene. The authors draw from a published database of eastern African ungulate communities spanning the last 7 Myr to examine temporal changes in various aspects of biodiversity and trait-environment relationships through time. The stated aim is to “evaluate whether the functional link between communities of herbivorous, eastern African megafauna and their environments (i.e., functional trait-environment relationships) was disrupted as biodiversity losses occurred over the past 7.4 Ma”

The authors show that long-term biodiversity loss occurred as grasslands expanded, and that between 1.7 Ma and the early Holocene, there is a disruption of trait-environment relationships—i.e., the trait makeup of faunal communities provide a signal that is woodier than estimates of woody cover derived from the $\delta^{13}\text{C}$ of carbonates. It's not entirely clear what to make of this “disruption of community ecological function” – why would the traits not align with the environment for an interval of 1.7 Myr? The tentative suggestions offered in the text suggest that the authors don't know what to make of this either, but they repeatedly imply that the appearance *Homo erectus* and the development of Acheulean technology is to blame. The text is sufficiently weaselly (e.g., line 301: “it is possible that”; line 305: “they may have”; line 311: “further research is required”) that there is room for deniability, but it seems clear that hominin impacts are a focus—e.g., analytical decisions about cutoffs between bins are based on key events in hominin evolution and there is repeated mention of how the trend relates to the appearance of Acheulean technology and/or *Homo erectus* (e.g., lines 174-175; lines 276-277; line 309).

Overall, I think the exercise is interesting but I have reservations.

Interpretation:

How are we supposed to interpret the lengthy and persistent disruption of trait-environment relationships? This is the key observation of the study. My understanding is that rapid landscape transformation could lead to such disruptions (at least briefly), but that doesn't readily explain a trend spanning 1.7 Myr. More importantly—why are hominin impacts even on the table as a potential driver? I see no citations to literature showing that a lengthy and persistent mismatch between traits and the environment is expected from anthropogenic impacts (if such a literature exists). It's worth pointing out a recent ecometrics paper (Short et al. 2023 in PNAS) that includes co-authors of this study. Short et al. (2023) made the case that equilibrium between traits and the environment happens very quickly relative to evolutionary timescales. To reinforce this idea, they pointed to the dramatic losses of megafauna near the end of the Pleistocene: “Even the Pleistocene megafaunal extinctions, which removed many large-bodied species from mammal communities on most continents, did not cause a large or persistent mismatch.” There is a strong case to be made that humans played a decisive role in the end-Pleistocene megafaunal extinctions, when the majority of Earth's megafauna were wiped out. If trait-environment relationships were not disrupted by these catastrophic losses, then what makes us think that *Homo erectus* is capable of fundamentally altering the function of Pleistocene faunal communities? And why would present-day African communities, which have unquestionably been influenced by recent human impacts, show a robust ecometric fit with the environment? (or is it just because these modern systems are used in the training set?) Hominin impacts therefore seem quite unlikely, but the authors don't offer any obvious alternatives—except a brief and tentative speculation about pulses of aridity in the past 1 Myr.

Analytical decisions:

1. The bins used to address ecometric anomalies are unusual. The boundaries between bins correspond with a somewhat random assortment of climatic (e.g., mid-Pleistocene transition) or archaeological events (e.g., development of Acheulean technology) of interest. Some bins span millions of years (e.g., 7.4 to 5.0 Ma) while others are much shorter (e.g., 1.7 to 1.0 Ma). This is a

curious analytical decision. Would a more typical binning strategy (e.g., uniform bins) lead to the same outcome? There are a few problems with the binning strategy used here. First, the archaeological events used to define the bins are all subject to change pending further research effort—e.g., the Oldowan was just pushed back a few hundred thousand years so 2.6 Ma no longer works. We shouldn't be surprised if the Acheulean were eventually pushed back in time as well. Second, all of these bins are much longer than the 250 kyr bins used in the biodiversity analysis shown in Figure 1, which makes it near impossible to compare one to the other. Third, these non-uniform bins also make it difficult to think about long-term trends, and how they might relate to environmental and ecological changes not considered by the authors. Why not use the same 250 kyr bins used in the biodiversity analysis? Or even better, why bin the data at all?

2. For 21 of the 79 sites examined here, there are no paleosol $\delta^{13}C$ data that can be used to estimate woody cover. The authors note that in these cases they have "obtained their mean woody cover estimates directly from the literature." Looking through the supplementary data (Dataset S3) it becomes clear that the authors ballparked various woody cover estimates from qualitative descriptions of the paleoenvironment (e.g., derived from faunal composition). I see why you'd want to flesh out your dataset as much as possible, but I don't think this is a good idea. These qualitative descriptions are super vague and imprecise. To provide an example, many researchers have compared fossil bovid communities with those found in Africa today to provide a general indication of the paleoenvironment (this goes back to important work by Vrba, 1980). Many of the modern communities dominated by Antilopini (e.g., gazelles) and Alcelaphini (e.g., wildebeest) are often said to represent open grasslands, and fossil communities that look similar (e.g., lots of Antilopini and Alcelaphini) are said to sample open grasslands. However, many of the modern "open grassland" ecosystems are much woodier than our qualitative descriptions would imply (e.g., Masai Mara, 61% woody cover; Omo National Park, 51% woody cover; Nairobi National Park, 56% woody cover). It wouldn't be unreasonable for a fossil community interpreted as sampling an open grassland to have anywhere from 0-60% woody cover. I admire the effort to try and put numbers on your fossil assemblages for which no direct estimates of woody cover are available, but they shouldn't be used for any sort of quantitative analysis. The authors have conducted separate analyses for the 58 sites with $\delta^{13}C$ data—these are the only analyses worth considering here. Ditch the ones with the ballpark estimates as they are unreliable.

Other comments:

Line 79-80: "The relative dearth of megafauna on Earth today dates to their dramatic declines throughout the Plio-Pleistocene" – throughout the Plio-Pleistocene? Most of the global losses are within the past 50,000 years. The next sentence makes this clear, but I'd rephrase this first sentence.

Figure 1: It would be easier to see the trends in woody cover (panel A) if you rescaled the y-axis. Maybe 0 to 0.75?

Line 140-142: "During the terminal Miocene before ~5 Ma, grasslands expanded (Figure 1, left-most blue region) and formed a mosaic environment with woodlands." I recommend being very careful/precise with your descriptions here. How are you defining your vegetation types (e.g., grassland, woodland). And at what spatial scale are you referring to? – is all of eastern Africa a mosaic environment? Probably not but that's what the text seems to imply. Paleosol carbonate $\delta^{13}C$ records show important differences between localities (e.g., Afar versus Turkana basin; West Turkana versus Shungura), as well as within them.

Line 151-152: "However, the decline began much earlier than documented losses in the functional diversity of large African carnivorans beginning ~2 Ma" But note that Werdelin and Lewis (2013) only looked at the past 3.5 Ma. Who knows what's going on with carnivoran functional diversity over the past 7 Ma?

Line 161-162: "The decline in woody cover (Figure 1a) likely occurred as grassland expansion intensified" – the decline in woody cover and grassland expansion are the same thing. The $\delta^{13}C$

record used to estimate woody cover is ultimately telling is about C3 versus C4 biomass. More C3 biomass in the present-day tropics (where there are no C3 grasses) is usually associated with more woody cover. More C4 biomass indicates more grasses.

Line 162-163: "...a force that may in turn have driven decreases in richness by extirpating species less adapted to grassland environments." Speculative post-hoc explanation. Had you observed the opposite trend, you could have instead suggested that grassland expansion drove an increase in richness as grassland lineages radiated.

Line 166-158: "Regardless, the broader similarities in these two trends reflect the positive relationship that has existed between mammalian biodiversity and primary productivity over the last 20 Ma 62." I'm missing something here—how are you inferring primary productivity? Why is this relevant?

Lines 170-172: "Losses in phylogenetic diversity...were the last to occur and were coincident with both environmental change and hominin evolution." Please clarify what you mean when you say coincident with environmental change and hominin evolution. Given the spatial and temporal focus of your paper, literally anything you could possibly observe is coincident with environmental changes and hominin evolution.

Line 173: The Oldowan just got older: <https://www.science.org/doi/full/10.1126/science.abo7452>

Line 180: "It emerged at a time in which woody cover decreased to <30%" Once again there is need to be precise with your description. Across what spatial scale does <30% woody cover apply? There are places that are dominated by C4 vegetation when *Homo erectus* emerges, but also places that are dominated by C3 vegetation. What does <30% apply to? It's not like all of eastern Africa was <30% woody cover, but that's what the text seems to imply. See also Line 242.

Line 259: change "consisted of" to "included" –the former reads as if this giraffe is the only thing present.

Line 290: *Camelus* is not the best example to use here. Isotopic and microwear evidence indicate that eastern African fossil *Camelus* was a mixed feeder preferring C3 browse (Cerling et al., 2015; Rowan et al., 2019). In terms of its actual diet (because not all hypsodont taxa are grazers), its removal is not at odds with a decline in C3 cover. I'd look for a better example.

Breakpoint analysis: Before reading too much into the breakpoints, it would be useful to examine whether the segmented regressions with one breakpoint are preferred to a linear model without a breakpoint. We'd hope for lower AIC scores in the segmented model.

Dataset S1: I see you have 1 loph for *Praedamalis* sp. and *P. deturi*. It's definitely 2 like other bovids. Worth double-checking your data.

Cerling, T.E., Andanje, S.A., Blumenthal, S.A., Brown, F.H., Chritz, K.L., Harris, J.M., Hart, J.A., Kirera, F.M., Kaleme, P., Leakey, L.N., Leakey, M.G., Levin, N.E., Manthi, F.K., Passey, B.H., Uno, K.T., 2015. Dietary changes of large herbivores in the Turkana Basin, Kenya from 4 to 1 Ma. *Proceedings of the National Academy of Sciences of the USA* 112, 11467-11472.

Levin, N., 2013. Compilation of East Africa Soil Carbonate Stable Isotope Data. Available at [dx.doi.org/10.1594/ieda/100231](https://doi.org/10.1594/ieda/100231).

Rowan, J., Martini, P., Likius, A., Merceron, G., Boisserie, J., 2019. New Pliocene remains of *Camelus grattardi* (Mammalia, Camelidae) from the Shungura Formation, Lower Omo Valley, Ethiopia, and the evolution of African camels. *Historical Biology* 31, 1123-1134.

Vrba, E.S., 1980. The significance of bovid remains as an indicator of environment and predation patterns, in: Behrensmeyer, A.K., Hill, A.P. (Eds.), *Fossils in the Making*. University of Chicago Press, Chicago, pp. 247-272.

Werdelin, L., Lewis, M.E., 2013. Temporal Change in Functional Richness and Evenness in the Eastern African Plio-Pleistocene Carnivoran Guild. PLoS One 8, e57944.

Reviewer #2:

Remarks to the Author:

This is a very interesting paper that deals on the disruption of trait-environment relationships in African megafauna that occurred during late Early Pleistocene to Middle Pleistocene times. I have enjoyed it and only have some minor contents for the authors, including:

- 1) It would be interesting to comment on the inverse relationship between the degree of megafaunal extinctions caused by human impact (overhunting and environmental deterioration) and the timing of human dispersal in the continents and islands (i.e., that most losses in biodiversity took place when humans arrived in recent times (for example America and New Zealand) compared to Africa, where humans coevolved with megafauna during more than two million years.
- 2) The development of hypsodont (i.e., high crowned) teeth is more closely related in ungulates to dwelling in open habitats, where the grass accumulates external abrasives (dust and grit), than to grazing due to the internal silicophytoliths of grasses (see Mendoza & Palmqvist 2008 in Journal of Zoology).
- 3) Monogastric, hindgut fermenters like perissodactyls are in disadvantage against ruminants (foregut fermenters) when grasses of high digestibility are available. However, if the limiting factor is food quality, not quantity (i.e., if foraging on high fiber grass of low quality value), non-ruminant herbivores are in advantage. There are many papers of Christine M. Janis dealing on this.
- 4) The 44-kg threshold for identifying megafaunal species can be problematic. For many researchers, megafauna comprises species weighing above 1000 kg (i.e., only very large mammals). Have these species been identified separately from other large mammals?
- 5) Do the categories (i.e., discretized values) of 1, 2 and 3 for hypsodonty roughly correspond to brachyodont, mesodont and hypsodont taxa?
- 6) There are examples of recent studies (e.g., Figueirido et al. 2019 in PNAS) focused on the temporal dynamics of ecotypes that are not only based on tooth ecometrics and body mass of ungulates, but also incorporate locomotor types, which are also related to the fraction of woody cover. Moreover, the study of Figueirido includes also carnivoran taxa, which provides a more accurate view of paleocommunities. This should be commented and discussed on the possibilities offered by such studies in order to characterize changes in habitat types.

REVIEWER COMMENTS

Reviewer #1 (Remarks to the Author):

This paper addresses biodiversity loss of eastern Africa megafauna through the Plio-Pleistocene. The authors draw from a published database of eastern African ungulate communities spanning the last 7 Myr to examine temporal changes in various aspects of biodiversity and trait-environment relationships through time. The stated aim is to “evaluate whether the functional link between communities of herbivorous, eastern African megafauna and their environments (i.e., functional trait-environment relationships) was disrupted as biodiversity losses occurred over the past 7.4 Ma”

Thank you very much for taking the time to review our manuscript and for your insightful comments. When we refer to line numbers below, we are referring to the line numbers in the tracked-changes version of the manuscript.

The authors show that long-term biodiversity loss occurred as grasslands expanded, and that between 1.7 Ma and the early Holocene, there is a disruption of trait-environment relationships—i.e., the trait makeup of faunal communities provide a signal that is woodier than estimates of woody cover derived from the $\delta^{13}\text{C}$ of carbonates. It’s not entirely clear what to make of this “disruption of community ecological function” – why would the traits not align with the environment for an interval of 1.7 Myr? The tentative suggestions offered in the text suggest that the authors don’t know what to make of this either, but they repeatedly imply that the appearance *Homo erectus* and the development of Acheulean technology is to blame. The text is sufficiently weaselly (e.g., line 301: “it is possible that”; line 305: “they may have”; line 311: “further research is required”) that there is room for deniability, but it seems clear that hominin impacts are a focus—e.g., analytical decisions about cutoffs between bins are based on key events in hominin evolution and there is repeated mention of how the trend relates to the appearance of Acheulean technology and/or *Homo erectus* (e.g., lines 174-175; lines 276-277; line 309).

We appreciate this insight. It is apparent that we could be clearer about the proximate and ultimate drivers of the observed disruption in trait-environment relationships. The proximate cause of trait-environment disruption is that the taxa that are well-adapted to the environmental conditions of specific sites are excluded from those sites. This proximate cause alone is sufficient to portray the main message of our manuscript, which is that only the biodiversity losses occurring since the mid-Pleistocene were associated with a disturbance to the ecological function of megafaunal communities. In other words, the fact that we observe an ecometric disruption is noteworthy, independent of its ultimate cause. However, we think you are right that a more thoughtful and careful discussion of the ultimate cause is warranted. We have modified our analyses and interpretations to more neutrally explore the potential influences of climate and hominins on our observed ecometric patterns (e.g., we no longer use analytical cutoffs that only pertain to hominin evolution). We have also done our best to strengthen our discussion so that we more thoroughly address potential ultimate causes of a long-term disruption of trait-environment relationships. See our response to your “Interpretation” comment for details.

Overall, I think the exercise is interesting but I have reservations.

We hope that our comments and revisions will help to alleviate these reservations.

Interpretation:

How are we supposed to interpret the lengthy and persistent disruption of trait-environment relationships? This is the key observation of the study. My understanding is that rapid landscape transformation could lead to such disruptions (at least briefly), but that doesn't readily explain a trend spanning 1.7 Myr. More importantly—why are hominin impacts even on the table as a potential driver? I see no citations to literature showing that a lengthy and persistent mismatch between traits and the environment is expected from anthropogenic impacts (if such a literature exists). It's worth pointing out a recent ecometrics paper (Short et al. 2023 in PNAS) that includes co-authors of this study. Short et al. (2023) made the case that equilibrium between traits and the environment happens very quickly relative to evolutionary timescales. To reinforce this idea, they pointed to the dramatic losses of megafauna near the end of the Pleistocene: “Even the Pleistocene megafaunal extinctions, which removed many large-bodied species from mammal communities on most continents, did not cause a large or persistent mismatch.” There is a strong case to be made that humans played a decisive role in the end-Pleistocene megafaunal extinctions, when the majority of Earth's megafauna were wiped out. If trait-environment relationships were not disrupted by these catastrophic losses, then what makes us think that *Homo erectus* is capable of fundamentally altering the function of Pleistocene faunal communities? And why would present-day African communities, which have unquestionably been influenced by recent human impacts, show a robust ecometric fit with the environment? (or is it just because these modern systems are used in the training set?) Hominin impacts therefore seem quite unlikely, but the authors don't offer any obvious alternatives—except a brief and tentative speculation about pulses of aridity in the past 1 Myr.

Thank you for this comment. We have now expanded our discussion of the climate mechanism we think is responsible for serving as a more plausible cause of the disruption in trait-environment relationships (see the final paragraph before the Conclusions section).

In the case of Short et al. (2023), the authors examined a single locomotor trait in North American taxa. The end-Pleistocene extinctions, while having an extreme effect on the composition of North American communities, did not result in the overall decimation locomotor traits, per se. As a result, the surviving species were able to re-equilibrate quickly. The same cannot necessarily be said of the functional traits that we examined here in eastern Africa.

We believe a strong case can be made that climatic patterns (specifically, periods of aridity) were a cause of the long-term ecometric disruption that we observe in eastern Africa. Faith et al. (2019) describes how pulses of aridity after 1 Ma led to “substantial shifts in community composition after ~1 Ma that cannot be explained by grassland expansion”. They find that while grasslands continued to expand (Figure 4A in their paper), grazers surprisingly underwent a decline in richness (Figure 2 in their paper). They explain how aridity may have elicited such a decline. Their findings align nicely with ours: a long-term decrease in grazer richness, coupled with expanding grasslands, would lead to herbivore communities whose traits persistently predict

environments that are woodier than they truly were (leading to our observed shift towards persistent negative ecometric anomalies – see **Figure 2**). Of course, Faith et al. (2019) focuses their discussion on the past 1 Ma, and our observed shift begins before then. However, prior research (e.g., Trauth et al. (2009) & Maslin et al. (2014)) shows that climate variability and aridity began to increase ~1.9 Ma, which aligns more closely with our results. Indeed, a closer look at Figure 2 in Faith et al. (2019) indicates that grazer richness began to plateau and slightly decline even prior to 1 Ma. Put together, we believe that increasing periods of aridity may provide a viable explanation of our observed long-term shift. We expand on this idea in the main text (see the final paragraph before the Conclusions section).

In addition, we reduce our analytical and interpretative focus on hominins as follows:

- As per your suggestion below, we now consider the emergence of Oldowan tools to occur at 3 Ma (instead of 2.6 Ma). Consequently, the Oldowan alone is no longer used as an independent cutoff point between ecometric bins, as it is grouped together with the mid-Pliocene warm period (see **Figure 1-3, Figure S8**).
- We now consider the increase in climate variability and aridity in conjunction with hominin events between 1.9 to 1.7 Ma. Thus, the development of Acheulean technology at 1.7 Ma is also no longer an independent cutoff point between ecometric bins.
- We point out why it is unlikely that hominin impacts resulted in our observed ecometric shift (see lines 407-409).

References:

Faith, J. T., Rowan, J., & Du, A. (2019). Early hominins evolved within non-analog ecosystems. *Proceedings of the National Academy of Sciences*, 116(43), 21478-21483.

Maslin, M. A., Brierley, C. M., Milner, A. M., Shultz, S., Trauth, M. H., & Wilson, K. E. (2014). East African climate pulses and early human evolution. *Quaternary Science Reviews*, 101, 1-17.

Trauth, M. H., Larrasoña, J. C., & Mudelsee, M. (2009). Trends, rhythms and events in Plio-Pleistocene African climate. *Quaternary science reviews*, 28(5-6), 399-411.

Analytical decisions:

1. The bins used to address ecometric anomalies are unusual. The boundaries between bins correspond with a somewhat random assortment of climatic (e.g., mid-Pleistocene transition) or archaeological events (e.g., development of Acheulean technology) of interest. Some bins span millions of years (e.g., 7.4 to 5.0 Ma) while others are much shorter (e.g., 1.7 to 1.0 Ma). This is a curious analytical decision. Would a more typical binning strategy (e.g., uniform bins) lead to the same outcome? There are a few problems with the binning strategy used here. First, the archaeological events used to define the bins are all subject to change pending further research effort—e.g., the Oldowan was just pushed back a few hundred thousand years so 2.6 Ma no longer works. We shouldn't be surprised if the Acheulean were eventually pushed back in time as well. Second, all of these bins are much longer than the 250 kyr bins used in the biodiversity analysis shown in Figure 1, which makes it near impossible to compare one to the other. Third, these non-uniform bins also make it difficult to think about long-term trends, and how they might

relate to environmental and ecological changes not considered by the authors. Why not use the same 250 kyr bins used in the biodiversity analysis? Or even better, why bin the data at all?

Your concerns about our binning strategy are fair – we appreciate your thoughts on them. We hope our comments and edits here are helpful in addressing them.

The advantage of our binning/violin plot approach is that it allows us to use a statistical framework to assess when trait-environment relationships were disrupted. The goal of our analysis in **Figure 2** is not to identify raw trends (as in **Figure 1**), but instead to broadly determine when a shift from a baseline ecometric anomaly of 0 occurred. Producing a trend (e.g., a LOESS regression curve) of anomalies without bins would not be appropriate for this task for two reasons: 1) more sites occur at more recent times, and a raw trend would not be able to account for that temporal sampling bias. Without a binning and bootstrapping approach (like we use in **Figure 2**), sites that occur further back in time would have an artificially greater influence than those that are more recent. 2) A lack of bins (and associated violin plots) would make it more challenging to statistically determine when anomalies are equivalent to 0, and alternatively when they deviate from 0.

Unfortunately, due to sample size restrictions, 250-ka bins are not appropriate for **Figure 2**. Such small time bins would contain too few sites to produce interpretable violin plots. We would not be able to effectively perform statistical analyses on the bins' distributions of values. Using 250-ka bins is more appropriate in **Figure 1** where the goal is instead to identify raw trends, and where the distribution of values in each individual bin is not the focus. 250-ka bins are especially appropriate for our biodiversity curves: there are more species than sites, and thus the sample sizes of species in each 250-ka bin are greater than the sample sizes of sites. Cumulatively, this explains why we use much larger time bins for sites in **Figure 2**.

We understand the concerns about our larger and irregularly sized time bins, and we perform further analyses regarding them (see below), but we also believe they remain appropriate for our purposes. Our goal with these bins is to determine how ecometric relationships generally behaved during/after key periods that were significant both to environmental change and to hominin evolution (see below and see our revision for our updated periods). We therefore see virtue in structuring our bins based on the timings of these periods. While we may not be able to comment on exact points in time, we can still draw conclusions on the overall behavior of ecometric relationships through time. Further, we believe our ecometric results are comparable to those in **Figure 1**, despite the different approach used there. **Figure 1** tells us that biodiversity losses occurred at multiple points throughout the Plio-Pleistocene, and **Figure 2** tells us that only some of those losses were broadly coincident with ecometric disruption.

With that said, we have performed the following analyses and modifications to address your concerns:

- We now consider the Oldowan to have occurred at 3 Ma and not 2.6 Ma. Accordingly, we have modified the bins in **Figure 2**, **S6**, **S7**, **S8**, and **S10**.
- In the main text, we have avoided using time bins that differ drastically in size (see **Figure 2**). However, we have included a more fine-focused binning structure in the

supplements to show how ecometric anomalies behaved within the events occurring 3.3 to 3 Ma and 1.9 to 1.7 Ma (see **Figure S8**).

- Our ecometric time bins (as well as our discussion of our biodiversity curves) are now structured around intervals of time that played central roles in both environmental change and hominin evolution. Each of 7.4 to 5 Ma, 3.3 to 3 Ma, 1.9 to 1.7 Ma, and ≥ 1 Ma (see **Figure 1, S13**) encompass both key environmental and hominin events. Thus, our ecometric time bins no longer correspond with a random assortment of times and events.
- We have repeated our ecometric analysis using bins of more uniform size. Using exactly uniform bins while also including a meaningful number of bins is a challenge and not the most appropriate, given the substantially smaller sample of sites occurring at earlier times. This is particularly true after we reduce our fossil site count from 79 to 58, based on the comment below (for example, only 8 sites have age ranges that overlap the first 3 Ma of our time series). Thus, we have made our first two bins 2 Ma in size and our next two bins slightly smaller. These bins are still considerably more uniform in size relative to our prior analyses (see **Figure S9**).

Overall, our modifications and analyses have upheld our current findings (undisturbed ecometric relationships in the Pliocene and early Pleistocene, followed by a disturbance). We hope that they help to demonstrate the utility of our approach and the validity of our results.

2. For 21 of the 79 sites examined here, there are no paleosol $\delta^{13}\text{C}$ data that can be used to estimate woody cover. The authors note that in these cases they have “obtained their mean woody cover estimates directly from the literature.” Looking through the supplementary data (Dataset S3) it becomes clear that the authors ballparked various woody cover estimates from qualitative descriptions of the paleoenvironment (e.g., derived from faunal composition). I see why you’d want to flesh out your dataset as much as possible, but I don’t think this is a good idea. These qualitative descriptions are super vague and imprecise. To provide an example, many researchers have compared fossil bovid communities with those found in Africa today to provide a general indication of the paleoenvironment (this goes back to important work by Vrba, 1980). Many of the modern communities dominated by Antilopini (e.g., gazelles) and Alcelaphini (e.g., wildebeest) are often said to represent open grasslands, and fossil communities that look similar (e.g., lots of Antilopini and Alcelaphini) are said to sample open grasslands. However, many of the modern “open grassland” ecosystems are much woodier than our qualitative descriptions would imply (e.g., Masai Mara, 61% woody cover; Omo National Park, 51% woody cover; Nairobi National Park, 56% woody cover). It wouldn’t be unreasonable for a fossil community interpreted as sampling an open grassland to have anywhere from 0-60% woody cover. I admire the effort to try and put numbers on your fossil assemblages for which no direct estimates of woody cover are available, but they shouldn’t be used for any sort of quantitative analysis. The authors have conducted separate analyses for the 58 sites with $\delta^{13}\text{C}$ data—these are the only analyses worth considering here. Ditch the ones with the ballpark estimates as they are unreliable.

This is a fair point. Our analyses now exclude the 21 sites lacking quantitative estimates of woody cover. Consequently, we now only include 203 fossil megafauna species (as opposed to 223) in our analyses, because we have removed the species that only occurred at those 21 sites. Similarly, we now also only include 48 modern megafauna species (as opposed to 53), as we

have removed the species that only occur at the sites without available woody cover data (see **Dataset S4** for the 128 out of 203 sites with such data available). This has had only minor influences on our biodiversity (**Figure 1,3**) and ecometric results (**Figure 2**), but we figured it is worth noting.

Other comments:

Line 79-80: “The relative dearth of megafauna on Earth today dates to their dramatic declines throughout the Plio-Pleistocene” – throughout the Plio-Pleistocene? Most of the global losses are within the past 50,000 years. The next sentence makes this clear, but I’d rephrase this first sentence.

We have rephrased this sentence make it clearer (see lines 79-80).

Figure 1: It would be easier to see the trends in woody cover (panel A) if you rescaled the y-axis. Maybe 0 to 0.75?

Done.

Line 140-142: “During the terminal Miocene before ~5 Ma, grasslands expanded (Figure 1, left-most blue region) and formed a mosaic environment with woodlands.” I recommend being very careful/precise with your descriptions here. How are you defining your vegetation types (e.g., grassland, woodland). And at what spatial scale are you referring to? – is all of eastern Africa a mosaic environment? Probably not but that’s what the text seems to imply. Paleosol carbonate $\delta^{13}\text{C}$ records show important differences between localities (e.g., Afar versus Turkana basin; West Turkana versus Shungura), as well as within them.

We have rephrased this sentence to remove the vague mention of a “mosaic environment”, and we have combined it with the subsequent sentence to make it clear that we are only commenting on the specific localities in our dataset, as opposed to all of eastern Africa (see lines 155-157).

Line 151-152: “However, the decline began much earlier than documented losses in the functional diversity of large African carnivorans beginning ~2 Ma” But note that Werdelin and Lewis (2013) only looked at the past 3.5 Ma. Who knows what’s going on with carnivoran functional diversity over the past 7 Ma?

That is a good call. Given that this was more of a side point, we have decided to remove it from our discussion.

Line 161-162: “The decline in woody cover (Figure 1a) likely occurred as grassland expansion intensified” – the decline in woody cover and grassland expansion are the same thing. The $\delta^{13}\text{C}$ record used to estimate woody cover is ultimately telling is about C3 versus C4 biomass. More C3 biomass in the present-day tropics (where there are no C3 grasses) is usually associated with more woody cover. More C4 biomass indicates more grasses.

We have rephrased this sentence to strengthen the association between woody cover decline and grassland expansion (see lines 175-176).

Line 162-163: "...a force that may in turn have driven decreases in richness by extirpating species less adapted to grassland environments." Speculative post-hoc explanation. Had you observed the opposite trend, you could have instead suggested that grassland expansion drove an increase in richness as grassland lineages radiated.

We have added evidence from the literature to back up this claim, so that it does not remain a speculation (see line 177). Bobe (2006) found evidence of an overall decrease in the richness of large eastern African herbivores as grasslands expanded across the Plio-Pleistocene, and Faith et al. (2019) found concurrent decreases in the richness of browsers and mixed feeders specifically.

References:

Bobe, R. (2006). The evolution of arid ecosystems in eastern Africa. *Journal of Arid Environments*, 66(3), 564-584.

Faith, J. T., Rowan, J., & Du, A. (2019). Early hominins evolved within non-analog ecosystems. *Proceedings of the National Academy of Sciences*, 116(43), 21478-21483.

Line 166-158: "Regardless, the broader similarities in these two trends reflect the positive relationship that has existed between mammalian biodiversity and primary productivity over the last 20 Ma 62." I'm missing something here—how are you inferring primary productivity? Why is this relevant?

This was meant as more of a side point, and as such we have removed it.

Lines 170-172: "Losses in phylogenetic diversity... were the last to occur and were coincident with both environmental change and hominin evolution." Please clarify what you mean when you say coincident with environmental change and hominin evolution. Given the spatial and temporal focus of your paper, literally anything you could possibly observe is coincident with environmental changes and hominin evolution.

We have modified this sentence to make it clearer and more specific (see lines 182-184).

Line 173: The Oldowan just got older:
<https://www.science.org/doi/full/10.1126/science.abo7452>

Thank you for bringing this to our attention. We have now accounted for this new Oldowan timing in our analyses (see **Figure 1-2**).

Line 180: "It emerged at a time in which woody cover decreased to <30%" Once again there is need to be precise with your description. Across what spatial scale does <30% woody cover apply? There are places that are dominated by C4 vegetation when *Homo erectus* emerges, but also places that are dominated by C3 vegetation. What does <30% apply to? It's not like all of eastern Africa was <30% woody cover, but that's what the text seems to imply. See also Line

242.

In both instances, we had mentioned (in the submitted manuscript) that woody cover decreased with respect to “the sites in which the large herbivores occurred (**Figure S2**)” (line 181 in the submitted manuscript) and “among the sites with herbivores (**Figure S2**)” (line 242). Our goal was to make it clear that these woody cover patterns apply only to the sites that we study, and not to eastern Africa in general. We have modified these two phrases to clarify and emphasize them (see lines 242 and 333).

Line 259: change “consisted of” to “included” –the former reads as if this giraffe is the only thing present.

Done (see line 350).

Line 290: Camelus is not the best example to use here. Isotopic and microwear evidence indicate that eastern African fossil Camelus was a mixed feeder preferring C3 browse (Cerling et al., 2015; Rowan et al., 2019). In terms of its actual diet (because not all hypsodont taxa are grazers), its removal is not at odds with a decline in C3 cover. I’d look for a better example.

Fair point. We have replaced Camelus with the genus Menelikia which, like other reduncin species, was a grazer (see line 394).

Breakpoint analysis: Before reading too much into the breakpoints, it would be useful to examine whether the segmented regressions with one breakpoint are preferred to a linear model without a breakpoint. We’d hope for lower AIC scores in the segmented model.

Thank you for mentioning this. We have performed a Davies test on all our breakpoint analyses using the ‘segmented’ package in R. The Davies test is appropriate for the purpose of determining whether a breakpoint is preferred over no breakpoint. We find that in all cases, a breakpoint is indeed preferred. We have included the details of this in the supplements (see lines 283-287 in the supplements).

Dataset S1: I see you have 1 loph for Praedamalis sp. and P. deturi. It’s definitely 2 like other bovids. Worth double-checking your data.

We have changed the loph count for these species to 2, and we have double-checked that our data are otherwise accurate.

Cerling, T.E., Andanje, S.A., Blumenthal, S.A., Brown, F.H., Chritz, K.L., Harris, J.M., Hart, J.A., Kirera, F.M., Kaleme, P., Leakey, L.N., Leakey, M.G., Levin, N.E., Manthi, F.K., Passey, B.H., Uno, K.T., 2015. Dietary changes of large herbivores in the Turkana Basin, Kenya from 4 to 1 Ma. Proceedings of the National Academy of Sciences of the USA 112, 11467-11472.

Levin, N., 2013. Compilation of East Africa Soil Carbonate Stable Isotope Data. Available at dx.doi.org/10.1594/ieda/100231.

Rowan, J., Martini, P., Likius, A., Merceron, G., Boisserie, J., 2019. New Pliocene remains of Camelus grattardi (Mammalia, Camelidae) from the Shungura Formation, Lower Omo Valley,

Ethiopia, and the evolution of African camels. *Historical Biology* 31, 1123-1134.

Vrba, E.S., 1980. The significance of bovid remains as an indicator of environment and predation patterns, in: Behrensmeyer, A.K., Hill, A.P. (Eds.), *Fossils in the Making*. University of Chicago Press, Chicago, pp. 247-272.

Werdelin, L., Lewis, M.E., 2013. Temporal Change in Functional Richness and Evenness in the Eastern African Plio-Pleistocene Carnivoran Guild. *PLoS One* 8, e57944.

Reviewer #2 (Remarks to the Author):

This is a very interesting paper that deals on the disruption of trait-environment relationships in African megafauna that occurred during late Early Pleistocene to Middle Pleistocene times. I have enjoyed it and only have some minor contents for the authors, including:

Thank you very much for taking the time to review our manuscript and for your insightful comments. When we refer to line numbers below, we are referring to the line numbers in the tracked-changes version of the manuscript.

1) It would be interesting to comment on the inverse relationship between the degree of megafaunal extinctions caused by human impact (overhunting and environmental deterioration) and the timing of human dispersal in the continents and islands (i.e., that most losses in biodiversity took place when humans arrived in recent times (for example America and New Zealand) compared to Africa, where humans coevolved with megafauna during more than two million years.

We agree that this is an interesting and worthwhile point, and we have included it as part of our introduction (see lines 86-87).

2) The development of hypsodont (i.e., high crowned) teeth is more closely related in ungulates to dwelling in open habitats, where the grass accumulates external abrasives (dust and grit), than to grazing due to the internal silicophytoliths of grasses (see Mendoza & Palmqvist 2008 in *Journal of Zoology*).

Thank you for pointing this out. We have modified our text to reflect this (see line 108 in the main text, as well as lines 108-109 and 396-397 in the supplements).

3) Monogastric, hindgut fermenters like perissodactyls are in disadvantage against ruminants (foregut fermenters) when grasses of high digestibility are available. However, if the limiting factor is food quality, not quantity (i.e., if foraging on high fiber grass of low quality value), non-ruminant herbivores are in advantage. There are many papers of Christine M. Janis dealing on this.

Thanks for mentioning this. We have incorporated the idea that when vegetation quantity (and not quality) is limited, non-ruminants are at a disadvantage. We have cited one of Christine Janis' papers in doing so (see lines 245-246).

4) The 44-kg threshold for identifying megafaunal species can be problematic. For many researchers, megafauna comprises species weighing above 1000 kg (i.e., only very large mammals). Have these species been identified separately from other large mammals?

This is a fair point. We used the 44-kg threshold because it is one of the most common (if not the most common) thresholds used to define mammalian megafauna in the literature (see Table S1 in Ripple et al. (2019)). Unfortunately, the sample size of species weighing above 1,000 kg in our dataset is limited, making it difficult to re-run all our analyses using just those species. However, to address the potential issue with the 44-kg threshold, we have repeated our ecometric analysis using a megafaunal threshold of 100 kg instead (a threshold that has been used for mammalian herbivores, as mentioned in the review of Moleon et al. (2020)). We find similar results using this threshold as well (see **Figure S10**).

References:

Moleon, M., Sánchez-Zapata, J. A., Donazar, J. A., Revilla, E., Martin-Lopez, B., Gutierrez-Canovas, C., ... & Tockner, K. (2020). Rethinking megafauna. *Proceedings of the Royal Society B*, 287(1922), 20192643.

Ripple, W. J., Wolf, C., Newsome, T. M., Betts, M. G., Ceballos, G., Courchamp, F., ... & Worm, B. (2019). Are we eating the world's megafauna to extinction?. *Conservation Letters*, 12(3), e12627.

5) Do the categories (i.e., discretized values) of 1, 2 and 3 for hypsodonty roughly correspond to brachyodont, mesodont and hypsodont taxa?

That is correct. We have clarified this point in the text (see line 525).

6) There are examples of recent studies (e.g., Figueirido et al. 2019 in PNAS) focused on the temporal dynamics of ecotypes that are not only based on tooth ecometrics and body mass of ungulates, but also incorporate locomotor types, which are also related to the fraction of woody cover. Moreover, the study of Figueirido includes also carnivoran taxa, which provides a more accurate view of paleocommunities. This should be commented and discussed on the possibilities offered by such studies in order to characterize changes in habitat types.

We agree that this is worth noting, and we have commented on it as part of our conclusions (see lines 486-488).

Reviewers' Comments:

Reviewer #1:

Remarks to the Author:

I appreciate the authors' efforts to address the concerns I raised during the previous review. My remaining edits are fairly minor. Line numbers below pertain to the version of the manuscript w/ track changes:

Line 151: Section header "Megafaunal Biodiversity Declined as Environments Changed and Hominins Evolved" – I made a similar comment last time around on a passage elsewhere in the manuscript, noting that literally anything that happened in Africa over the past 7 Myr is taking place as environments changed and hominins evolved (i.e., adding "as Environments Changed and Hominins Evolved" doesn't add much). Perhaps just shorten to "Megafaunal Biodiversity Decline"? Or "Megafaunal Biodiversity Decline and its Ecological and Paleoanthropological Context"?

Line 154: "Prior to ~8 Ma, woodlands and rainforests were prevalent..." I confess that my knowledge of things in the Miocene is rather hazy, but are we sure these are "rainforests" and not just "forests"?

Line 165-167: "Notably, the decline commenced well after hominins evolved ~7 Ma...and at a time in which no major turning points in hominin evolution occurred" Why is it notable that this happened well after hominins evolved? Is the implication that we should expect the decline in functional diversity to be synchronous with the emergence of the hominin clade? And if so why? If the goal is to evaluate hominin vs environmental impacts, then perhaps more notable is that the decline begins well before the appearance of any hominins plausibly capable of major ecosystem disruption.

Also, I see you've cited Brunet et al. here, but it seems increasingly unlikely that Sahelanthropus is a hominin (Macchiarelli et al., 2020). If needed, a safer citation for the emergence of the clade is probably the 10-7 Ma chimp-human split suggested by molecular evidence (Langergraber et al., 2012) rather than the controversial and spotty early hominin fossil record.

Langergraber, K., Prufer, K., Rowney, C., Boesch, C., Crockford, C., Fawcett, K., Inoue, E., Inoue-Muruyama, M., Mitani, J., Muller, M., Robbins, M., Schubert, G., Stoinski, T., Viola, B., Watts, D., Wittig, R., Wrangham, R., Zuberbuhler, K., Paabo, S., Vigilant, L., 2012. Generation times in wild chimpanzees and gorillas suggest earlier divergence times in great ape and human evolution. *Proceedings of the National Academy of Sciences of the United States of America* 109, 15716-15721.
Macchiarelli, R., Bergeret-Medina, A., Marchi, D., Wood, B., 2020. Nature and relationships of *Sahelanthropus tchadensis*. *Journal of Human Evolution* 149.

Reviewer #2:

Remarks to the Author:

I see that the authors have addressed all the issues suggested in my review. As a result, the manuscript contents have been improved and I think that it can be now accepted for publication.

REVIEWER COMMENTS

Reviewer #1 (Remarks to the Author):

I appreciate the authors' efforts to address the concerns I raised during the previous review. My remaining edits are fairly minor. Line numbers below pertain to the version of the manuscript w/ track changes:

Thank you very much again for reviewing our manuscript, for your positive feedback, and for undoubtedly improving the quality of our work.

Line 151: Section header “Megafaunal Biodiversity Declined as Environments Changed and Hominins Evolved” – I made a similar comment last time around on a passage elsewhere in the manuscript, noting that literally anything that happened in Africa over the past 7 Myr is taking place as environments changed and hominins evolved (i.e., adding “as Environments Changed and Hominins Evolved” doesn't add much). Perhaps just shorten to “Megafaunal Biodiversity Decline”? Or “Megafaunal Biodiversity Decline and its Ecological and Paleoanthropological Context”?

This is a good point. We have gone with your second suggestion for this heading, except that we prefer to use the word “Environmental” instead of “Ecological” to reflect the focus of our language on past environmental changes (see line 138).

Line 154: “Prior to ~8 Ma, woodlands and rainforests were prevalent...” I confess that my knowledge of things in the Miocene is rather hazy, but are we sure these are “rainforests” and not just “forests”?

We have changed this word to just “forests” (see line 141).

Line 165-167: “Notably, the decline commenced well after hominins evolved ~7 Ma...and at a time in which no major turning points in hominin evolution occurred” Why is it notable that this happened well after hominins evolved? Is the implication that we should expect the decline in functional diversity to be synchronous with the emergence of the hominin clade? And if so why? If the goal is to evaluate hominin vs environmental impacts, then perhaps more notable is that the decline begins well before the appearance of any hominins plausibly capable of major ecosystem disruption.

Also, I see you've cited Brunet et al. here, but it seems increasingly unlikely that Sahelanthropus is a hominin (Macchiarelli et al., 2020). If needed, a safer citation for the emergence of the clade is probably the 10-7 Ma chimp-human split suggested by molecular evidence (Langergraber et al., 2012) rather than the controversial and spotty early hominin fossil record.

Thank you for pointing this out. Based on your comments, we have decided to remove the phrase about the decline in functional diversity commencing well after hominins evolved ~7 Ma, and now for simplicity, we only mention that the decline is not associated with any turning points in hominin evolution (see lines 152-154). We have also replaced the citations of Brunet et al. in

both the manuscript and supplements with your suggested citation of Langergraber et al. (see our comment on line 445 of the main text and line 179 of the supplements).

Langergraber, K., Prufer, K., Rowney, C., Boesch, C., Crockford, C., Fawcett, K., Inoue, E., Inoue-Muruyama, M., Mitani, J., Muller, M., Robbins, M., Schubert, G., Stoinski, T., Viola, B., Watts, D., Wittig, R., Wrangham, R., Zuberbuhler, K., Paabo, S., Vigilant, L., 2012. Generation times in wild chimpanzees and gorillas suggest earlier divergence times in great ape and human evolution. *Proceedings of the National Academy of Sciences of the United States of America* 109, 15716-15721.

Macchiarelli, R., Bergeret-Medina, A., Marchi, D., Wood, B., 2020. Nature and relationships of *Sahelanthropus tchadensis*. *Journal of Human Evolution* 149.

Reviewer #2 (Remarks to the Author):

I see that the authors have addressed all the issues suggested in my review. As a result, the manuscript contents have been improved and I think that it can be now accepted for publication.

Thank you very much again for reviewing our manuscript, for your positive feedback, and for undoubtedly improving the quality of our work.